# Comprehensive Evaluation of High-Resolution Satellite Precipitation Products over the Qinghai–Tibetan Plateau Using the New Ground Observation Network

Zhaofei Liu [1,2,3]

1   Institute of Geographic Sciences and Natural Resources Research, Chinese Academy of Sciences, Beijing 100101, China; zfliu@igsnrr.ac.cn
2   University of Chinese Academy of Sciences, Beijing 100049, China
3   Key Laboratory of Carrying Capacity Assessment for Resource and Environment, Ministry of Natural Resources, Beijing 101149, China

**Abstract:** Satellite precipitation products (SPPs) have been widely evaluated at regional scales. However, there have been few quantitative comprehensive evaluations of SPPs using multiple indices. Ten high-resolution SPPs were quantitatively and comprehensively evaluated from precipitation occurrence and series indices using an improved rank score (RS) method in the data-scarce Qinghai–Tibetan Plateau (QTP). The new observation network, along with a number of national basic stations, was applied for SPP evaluation to obtain more reliable results. The results showed that the GPM and MSWEP showed the strongest overall performance, with an RS value of 0.75. CHIRPS and GPM had the strongest performance at measuring precipitation occurrence (RS = 0.92) and series (RS = 0.75), respectively. The optimal SPPs varied in evaluation indices, but also concentrated in the MSWEP, GPM, and CHIRPS. The bias of SPPs was markedly in the QTP, with relative error generally between −80% and 80%. In general, most SPPs showed the ability to detect precipitation occurrence. However, the SPPs showed relatively weak performance at measuring precipitation series. The mean Kling–Gupta efficiency of all stations was <0.50 for each SPP. The SPPs showed better performance in monsoon-affected regions, which mainly include the Yangtze, Yellow, Nu–Salween, Lancang–Mekong, Yarlung Zangbo–Bramaputra, and Ganges river basins. Performance was relatively poor in the westerly circulation areas, which mainly include the Tarim, Indus, and QTP inland river basins. The performance of SPPs showed a seasonal pattern during the year for most occurrence indices. The performance of SPPs in different periods was opposite in different indices. Therefore, multiple indices representing different characteristics are recommended for the evaluation of SPPs to obtain a comprehensive evaluation result. Overall, SPP measurement over the QTP needs further improvement, especially with regard to measuring precipitation series. The proposed improved RS method can also potentially be applied for comprehensive evaluation of other products and models.

**Keywords:** comprehensive evaluation; satellite precipitation; precipitation event; multiple indices; rank score; Tibetan Plateau





## 1. Introduction

Precipitation is an important process in hydrology, meteorology, ecology, and agriculture. Accurately estimating precipitation is crucial for weather forecasting, hydrological modeling and water resource management [1,2]. Gauge observation is considered to be the most accurate method for precipitation measurement [3]. However, most precipitation gauge stations in China are concentrated in the eastern part of the country. Western China, a substantially large area, has very few or even none of these, particularly in the Qinghai–Tibetan Plateau (QTP). The sparse distribution of precipitation gauges and discontinuity in recording sequences often results in poor analysis of the spatial and temporal features

of precipitation [4]. For example, the daily precipitation was 23.5 mm and 229.6 mm at two stations located within 20 km of each other and was varied five times within a 10 km range [5]. Gauge precipitation is also underestimated by approximately 40% at the 6–7 km distance range [6].

Satellite precipitation estimates effectively solve the problems associated with insufficient spatial representation of precipitation, even in complex terrains, because they monitor area precipitation. In other words, satellite measurements directly represent spatial area precipitation. Recently, the satellite-based precipitation products (SPPs) are becoming more accurate and reliable, because they combine gauge observations and multi-satellite data. The increased availability and consistency and high spatiotemporal-resolution of satellite-based precipitation data make it widely applicable. Satellite sensors detect precipitation by measuring precipitation-related variables and developing inversion algorithms [7]. These sensors mainly include radars, LiDARs, and passive microwave radiometers [8]. LiDAR observations can even detect light rain events [9]. However, SPPs have inherent drawbacks associated with the indirect nature of the relationship between observations and precipitation, inadequate sampling, and deficient retrieval algorithms [10]. These uncertainties and inaccuracies are more in complex terrains due to low gauge densities [11]. Therefore, performing an evaluation of global SPPs before their application at regional scale is crucial [12].

Evaluation of SPPs showed that the integrated multi-satellite retrieval (IMERGs) products from the global precipitation measurements (GPMs) exhibited the strongest performance compared to other products at global and region scales [13]. These included Australia [14,15], Bangladesh [16], Central Asia [17], the Eastern Himalayas [10], Nigeria [18], Thailand [19], and Vietnam [20]. These also showed that IMERGs were the strongest SPPs in China [3,13,21–23]. However, their performance was poor in western China, especially in the complex mountainous and arid regions [22–24].

QTP is the highest and largest plateau on Earth, with an average altitude of >4100 m above sea level [25,26]. Measuring accurate precipitation is challenging at the QTP, due to complex terrain, insufficient gauge sites, and high spatial heterogeneity in precipitation [27]. The evaluation of SPPs here showed that their performance was worse when compared to that of other regions in China [22–24]. The climate prediction center morphing method (CMORPH) performed better than precipitation estimation from remotely sensed information using artificial neural networks (PERSIANN) and tropical rainfall measuring mission (TRMM) products in the QTP [28,29]. When the GPM IMERG products were available, the GPM showed strongest performance. The evaluation of six SPPs showed the GPM outperformed other datasets in QTP [30]. It overestimated precipitation amounts and showed better performance for high elevations. The GPM performed better than the climate hazards group infrared precipitation with station data (CHIRPS), PERSIANN, and TRMM products in eastern Himalaya, located in southern QTP [10]. However, the ensemble multi-satellite precipitation dataset using the dynamic Bayesian model averaging scheme and global satellite mapping of precipitation (GSMaP) performed better than GPM in the QTP [27]. However, previous studies have only used national stations of the China meteorological administration (CMA), sparsely distributed in the QTP. The national Tibetan plateau data center (NTPDC) developed a new precipitation observation network comparable to the number of CMA stations. This provides a new and wider ground observation network for evaluating SPPs in the QTP. At present, SPP evaluations based on the new observation network are lacking. In addition, although multiple indices have been used, previous studies were qualitative comprehensive evaluations, and there were few quantitative comprehensive evaluations using multiple indices in the QTP.

In this study, an improved rank score (RS) method was developed for the quantitative comprehensive evaluation of SPPs. Both the new observation network and national basic stations were used to obtain reliable comprehensive evaluation results for the data scarce QTP. The ability of ten high-resolution SPPs for detecting precipitation occurrence and measuring precipitation series was evaluated at each station. The performance of SPPs in

detecting precipitation occurrence was analyzed at different levels of precipitation. The spatial and temporal performances of SPPs were also evaluated. Finally, the SPPs were quantitatively and comprehensively evaluated using the improved RS method in the QTP.

## 2. Materials and Methods

### 2.1. Data

#### 2.1.1. Gauge Precipitation Data

Gauge precipitation data from 161 stations were used for this study, including the CMA and NTPDC stations. There are eleven large river basins located in the QTP. Observation stations are distributed through nine large river basins (Figure 1), including the Tarim, Gansu Corridor inland, QTP inland, Indus, Yarlung Zangbo–Bramaputra, Ganges, Nu–Salween, Lancang–Mekong, Yangtze, and Yellow river basins. The southern and eastern basins are affected by monsoon, whereas the western and northern basins are affected by westerly circulation. The gauge precipitation data from national stations were provided by the CMA, including data series from 1951–2018. The raw dataset consisted of 12 h (8 p.m.–8 a.m. and 8 a.m.–8 p.m.) and 24 h (8 p.m.–8 p.m.) precipitation observations in Beijing time. Raw data based on local time were converted to data based on Coordinated Universal Time (UTC) to make it comparable with the SPPs. The UTC is equal to Beijing time minus 8 h. There are 89 national stations in the QTP. The new observation network of precipitation dataset was provided by the NTPDC. It included 72 stations set up by researchers [31–78] (Table S1). These stations provided daily and hourly precipitation series. Hourly series data were converted to UTC to make them comparable with SPPs.

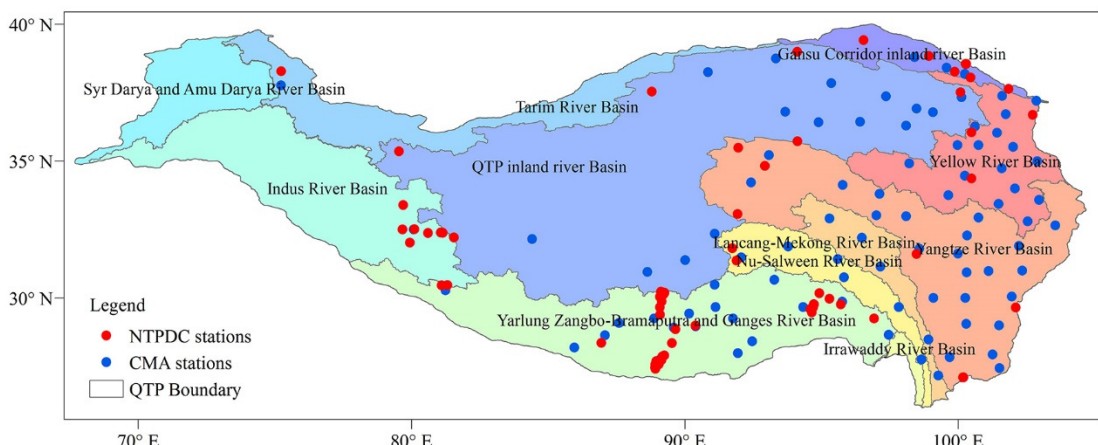

**Figure 1.** Location of observation stations in the QTP (Different colors represent different river basins).

#### 2.1.2. SPPs

Ten SPPs were used in this study (Table 1). CHIRPS is a quasi-global satellite precipitation dataset (https://chc.ucsb.edu/data/chirps (accessed on 28 January 2023)) [79]. The CMORPH is a bias-corrected product (https://www.ncei.noaa.gov/products/climate-data-records/precipitation-cmorph (accessed on 28 January 2023)) [80]. The GSMaP products were developed by the Japan aerospace exploration agency to achieve higher temporal and spatial resolutions using passive microwave sensors and infrared radiometers (https://developers.google.cn/earth-engine/datasets/catalog (accessed on 28 January 2023)) [81]. The GPM IMERG Final is a level 3 product (https://disc.gsfc.nasa.gov/datasets/ (accessed on 28 January 2023)) [82]. This product uses multiple passive microwave satellite sensors. It combines gauge data, microwave-calibrated infrared satellite estimates, and other precipitation estimators at a fine spatial resolution. The multi-source weighted-ensemble precipitation (MSWEP) product merges satellite, reanalysis, and gauge data (http://www.gloh2o.org/mswep/ (accessed on 28 January 2023)) [83]. The PERSIANN product is a high-resolution (up to 0.04°) product developed by the University of California, Irvine (http://chrsdata.eng.uci.edu/ (accessed on 28 January 2023)) [84]. There were four

PERSIANN products used in this study, including the initial satellite measurements, the dynamic infrared rain at near real time (PDIR-Now), the cloud classification system (CCS), and the climate data record (CDR) [84–86]. Hourly SPPs at UTC were converted to Beijing time make them comparable with gauge observations at daily NTPDC stations.

**Table 1.** SPPs datasets used in this study.

| Product | Version | Spatial Resolution | Temporal Resolution | Period | Abbreviation |
|---|---|---|---|---|---|
| CHIRPS | V2 | 0.05° × 0.05° | Daily | 1981.01–present | CHI |
| CMORPH | | 8 km × 8 km | 3-hourly | 1998.01–present | CMO |
| GSMaP | MVK V7 | 0.1° × 0.1° | hourly | 2014.03–present | GaM |
| | NRT V7 | 0.1° × 0.1° | hourly | 2000.03–present | GaN |
| GPM | IMERG-Final | 0.1° × 0.1° | 3-hourly | 2000.06–present | GPM |
| MSWEP | | 0.1° × 0.1° | 3-hourly | 1979.01–present | MS |
| PERSIANN | CCS | 0.04° × 0.04° | 3-hourly | 2003.01–present | PCS |
| | PDIR-Now | 0.04° × 0.04° | 3-hourly | 2000.03–present | PDI |
| | CDR | 0.25° × 0.25° | Daily | 2000.01–2020.12 | PDR |
| | Initial version | 0.25° × 0.25° | 3-hourly | 2000.03–present | PER |

*2.2. Methods*

2.2.1. Evaluation Criteria on Detecting Precipitation Occurrence

The joint distribution is usually used to evaluate a satellite's ability to detect precipitation occurrence [87]. The marginal distribution of gauge observation and satellite estimates could be defined as true positive (*TP*, observed precipitation correctly detected by the satellite), true negative (*TN*, no precipitation observed nor detected by the satellite), false positive (*FP*, observed precipitation not detected by the satellite), and false negative (*FN*, precipitation detected by the satellite but not observed). The performance statistics applied in this study include the accuracy (*A′*), critical success index (*CSI*, also called threat score), false alarm ratio (*FAR*), false positive rate (*FPR*, also called false alarm rate), and probability of detection (*POD*, also called true positive rate or hit rate) [87–89]. The statistics are particularly useful when the precipitation event occurs substantially less frequently rather than not occur at all [87], which is consistent with the actual precipitation occurrence at the QTP. Calculations for these are as follows:

$$A\prime = \frac{TP + TN}{TP + FP + FN + TN} \tag{1}$$

$$CSI = \frac{TP}{TP + FP + FN} \tag{2}$$

$$FAR = \frac{FN}{TP + FN} \tag{3}$$

$$FPR = \frac{FN}{FN + TN} \tag{4}$$

$$POD = \frac{TP}{TP + FP} \tag{5}$$

*CSI* is the fraction of observed and detected precipitation that was correctly detected; *FAR* gives the fraction of precipitation detections that were actually false alarms; *POD* is the fraction of precipitation occurrences that were correctly detected. The precipitation event is defined as daily precipitation >0.1 mm.

### 2.2.2. Evaluation Criteria for Simulating Precipitation Series

Evaluation criteria for simulating monthly precipitation series included Pearson's correlation coefficient (*CC*), index of agreement (*IOA*), Kling–Gupta efficiency (*KGE*), Nash-Sutcliffe efficiency (*NSE*), and relative error (*RE*) [90–93]. The *IOA* and *NSE* evaluate the goodness-of-fit of the simulated and observed data series. The *RE* represents bias deviation from the observed values. The *KGE* is an evaluation index that integrates the *CC*, *RE* and standard deviation. The evaluation criteria were calculated as follows:

$$RE = \frac{1}{n}\sum_{i=1}^{n}\frac{X_{mi} - X_{oi}}{X_{oi}}, \tag{6}$$

$$CC = \frac{\sum_{i=1}^{n}\left(X_{oi} - X_o\right)\left(X_{mi} - X_m\right)}{\sqrt{\sum_{i=1}^{n}\left(X_{oi} - X_o\right)^2 \sum_{i=1}^{n}\left(X_{mi} - X_m\right)^2}} \tag{7}$$

$$IOA = 1 - \frac{\sum_{i=1}^{n}(X_{mi} - X_{oi})^2}{\sum_{i=1}^{n}\left(\left|X_{mi} - X_o\right| + \left|X_{oi} - X_o\right|\right)^2} \tag{8}$$

$$KGE = 1 - \sqrt{(1 - CC)^2 + RE^2 + \left(1 - \frac{SD_m}{SD_o}\right)^2} \tag{9}$$

$$SD = \sqrt{\frac{1}{n-1}\sum_{i=1}^{n}\left(X_i - X\right)^2} \tag{10}$$

$$NSE = 1 - \frac{\sum_{i=1}^{n}(X_{mi} - X_{oi})^2}{\sum_{i=1}^{n}\left(X_{oi} - X_o\right)^2} \tag{11}$$

where $X_{mi}$ and $X_{oi}$ are the $i$th values of the satellite and observed precipitation time series, respectively; $n$ is the time series length; $X_m$ and $X_o$ are the mean of the satellite and observed values, respectively; $SD_m$ and $SD_o$ are the standard deviations of the satellite and observed precipitation, respectively.

### 2.2.3. Comprehensive Evaluation Method

The above is the evaluation of SPPs based on each index. The *RS*-based method comprehensively evaluates the performance of the model using multiple indices [94,95]. It is similar to the comprehensive Technique for Order Preference by Similarity to an Ideal Solution evaluation method applied in other research fields [96], and subjectively merges *RS* values into integers ranging from 0–9. An improved *RS* method was developed in this study. It objectively evaluated the comprehensive performance of a model by using the deviation degree between each model and the optimal model value in each index. The improved *RS* method was calculated as follows:

$$RS_i = \begin{cases} \frac{x_i - x_{min}}{x_{max} - x_{min}}, & positive\ indexes \\ 1 - \frac{x_i - x_{min}}{x_{max} - x_{min}}, & negative\ indexes \end{cases} \tag{12}$$

where $x_i$ is the index value of a satellite product; $x_{min}$ and $x_{max}$ are the minimum and maximum values of the index value in all SPPs; positive indices are the A′, CC, CSI, IOA, KGE, NSE, and POD; negative indices are the absolute value of RE, FAR, and FPR.

The RS was first calculated from the mean value of each index for each SPP. The total RS for a SPP was obtained by averaging the RS for each index used. The performance of SPPs in detecting precipitation occurrence and measuring precipitation series was evaluated using the occurrence (POD, FAR, FPR, CSI, and A′) and series indices (RE, CC, IOA, KGE,

and NSE), respectively. The SPP measurements were comprehensively evaluated using the RS of all 10 indices.

Ten SPPs were evaluated using the gauge-observed precipitation data from QTP and the raw spatial resolution of each SPP was used for the evaluation. The pixels closest to each ground station were selected for each SPP. The evaluation was based on a daily time series, and evaluation period was the time of intersection of each station and SPP.

## 3. Results

### 3.1. Performance of SPPs in Detecting Precipitation Occurrence

Figure 2 shows the performance of the 10 SPPs in detecting precipitation occurrence in the QTP. The mean POD of all SPPs across all stations was 0.66. MSWEP showed the strongest performance at detecting a fraction of precipitation occurrence, with the POD of most stations (151/161) exceeding 0.90. The mean and median POD values of MSWEP across all stations were 0.96 and 0.97, respectively. This might be due to higher precipitation frequency in MSWEP than that in other products. The PDIR-Now and PERSIANN-CDR also showed higher ability, with the POD being >0.60 at 94% (151/161) and 96% (155/161) of stations, respectively. The mean POD values of the two products were approximately 0.80. CHIRPS showed the weakest performance, with mean and median POD < 0.40.

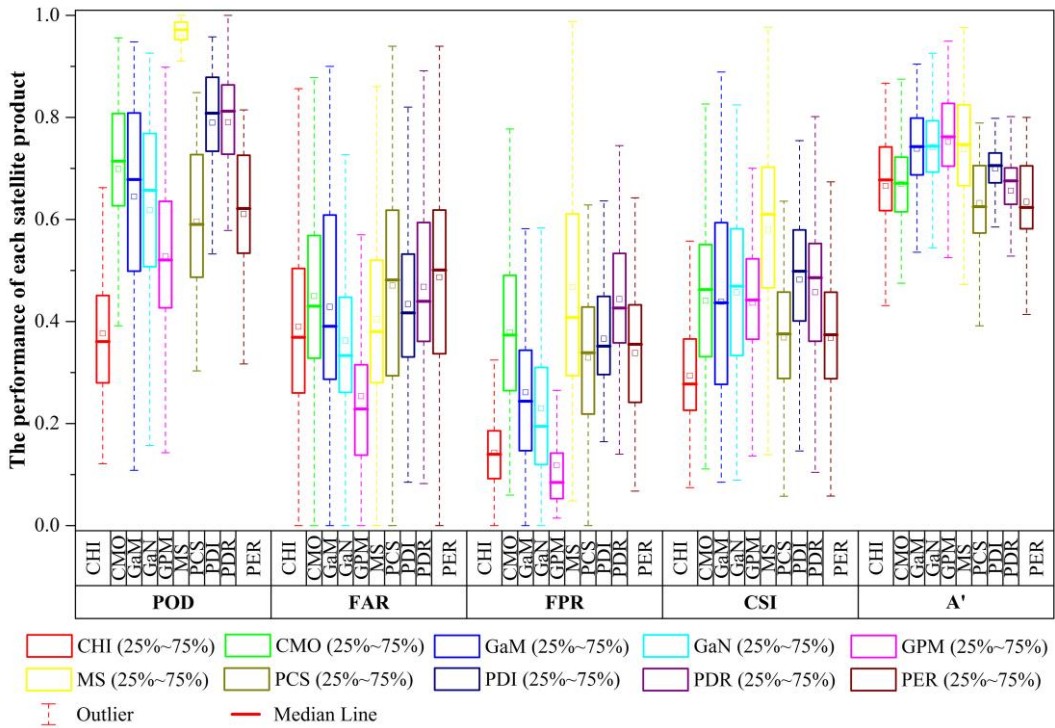

**Figure 2.** Performance of satellite products at detecting precipitation occurrence at the observed stations (the abbreviation of each product is shown in Table 1).

Although SPPs demonstrated the ability to capture precipitation occurrence, many false alarms of precipitation were present. FPR value was generally lower than FAR at the QTP. The mean FAR and FPR of all products across all stations were 0.41 and 0.31, respectively. GPM had the strongest performance in FAR, with <0.30 at 73% (118/161) of stations. The PERSIANN, PERSIANN-CCS, and PERSIANN-CDR showed greater FAR, with mean >0.45 for each product. GPM showed the strongest performance in FPR, with the mean value being 0.12. The FPR was <0.20 at 86% (139/161) of stations. CHIRPS also showed relatively better performance on FPR, with <0.20 at 91% (146/161) of stations.

The mean CSI of all SPPs was 0.43. The performance of each SPP varied substantially, with the median CSI of all stations ranging from 0.28–0.61. MSWEP had the strongest

performance, with the mean CSI being approximately 0.60. CSI was >0.50 at 71% (115/161) of stations. CHIRPS showed the weakest performance, with CSI < 0.30 at 60% (96/161) of stations.

The mean A′ of all SPPs was 0.69. The variation in performance of A′ was relatively small among the SPPs when compared with the other four occurrence indices. The GPM, MSWEP, GSMaP-NRT, and GSMaP-MVK showed better performance than other SPPs, with mean and median A′ of approximately 0.75. GPM had the strongest performance, with an A′ of 0.70 at 81% (130/161) of stations. The PERSIANN and PERSIANN-CCS showed the weakest performance, with a mean A′ of 0.63.

Figure 3 shows the performance of SPPs at different levels of precipitation. Taking 10 mm as the dividing line, the ability of SPPs to capture different levels of precipitation varied substantially. SPPs showed a strong ability to capture precipitation over 10 mm, with the mean POD > 0.60 in most products. The performance was relatively poor for the detection of precipitation <10 mm, with mean POD < 0.50 in most products. MSWEP showed the strongest performance at all levels, with mean POD > 0.90 at each level. The PDIR-Now and PERSIANN-CDR also showed a strong ability to capture precipitation <10 mm, with mean POD > 0.60. In addition to MSWEP, GSMaP-NRT and GSMaP-MVK also showed strong ability to capture rainstorm (daily precipitation > 50 mm), with a mean POD of approximately 0.80.

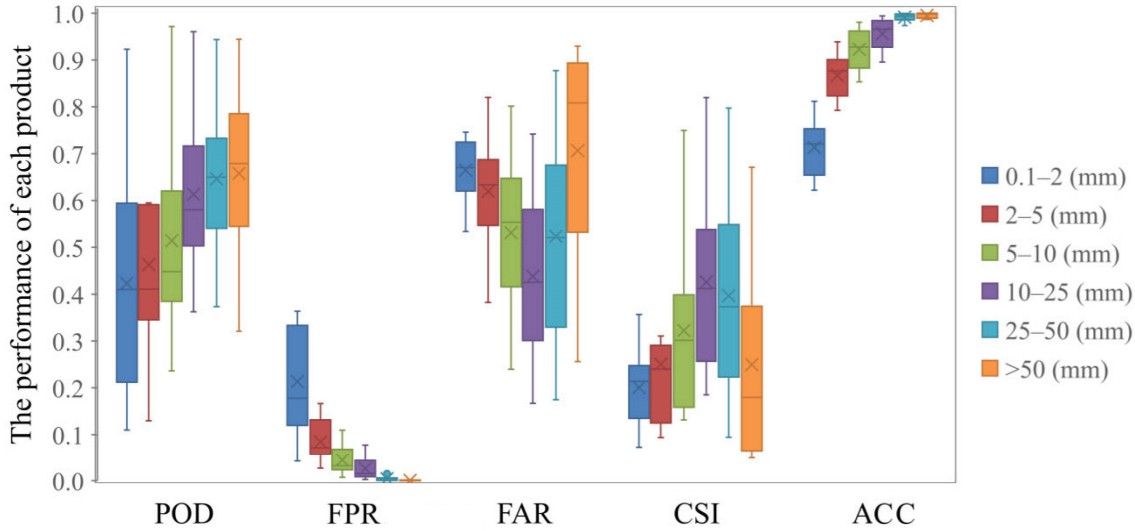

**Figure 3.** Performance of SPPs at different levels of precipitation amount.

In the A′-based evaluation, there was no substantial variation in the performance of satellite products at different levels of precipitation. Standard deviation estimated from the mean A′ of each SPP was within 0.05. The SPPs had the weakest performance at capturing precipitation occurrence at the level of 0.1–2 mm, with the median and mean A′ of all products being >0.70 and >0.60, respectively. All SPPs showed a strong ability to capture precipitation occurrence when precipitation was >10 mm, especially for heavy rain and rainstorm, with the mean A′ of each product ranging from 0.98 to 1.00. The evaluation results of the FPR were consistent with those of the A′ index.

In the CSI evaluation, SPPs had a stronger ability to capture precipitation occurrence at 10–50 mm than at other levels. All products had a weak ability to capture 0.1–2 mm precipitation, with the mean CSI of all stations being ≤0.40 for each product. MSWEP showed the strongest performance for precipitation ≤50 mm. In addition, the GPM and CMORPH showed better performance at the level of 10–50 mm, with the mean CSI being approximately 0.55. Other products had weak ability to capture precipitation <50 mm, with the mean CSI being generally <0.35. GPM showed the strongest performance at capturing rainstorm, with the mean CSI at 0.67. MSWEP also had better performance, with a mean

CSI of 0.62. However, other products showed poor ability to capture rainstorm, with the mean CSI of all stations being <0.30 for each product.

Overall, SPPs showed the ability to detect a fraction of precipitation occurrences, albeit with many false alarms, in the QTP. The MSWEP and GPM generally showed the strongest performance when evaluated on the basis of precipitation occurrence indices. In general, the performance of SPPs showed a positive correlation with precipitation levels in evaluation using the POD, FPR and A′ indices. With increasing precipitation amount in the annual cycle, the performance of SPPs first increased and then decreased when evaluated using the CSI and FAR indices. Satellite products had the strongest performance when capturing precipitation between 10 and 25 mm.

### 3.2. Spatial and Temporal Performance of SPPs

Figure 4 shows the spatial performance of SPPs at detecting precipitation occurrence. The ability of SPPs to capture precipitation occurrence varied substantially among stations with respect to the average performance of all products. SPPs had relatively strong ability to capture precipitation in the eastern (chiefly including the Yellow, Yangtze, Lancang–Mekong, and Nu–Salween river basins) and southern (the Ganges and Yarlung Zangbo–Bramaputra river basins) regions of the QTP, with the POD generally being >0.60. Performance was relatively poor at the source region of the Tarim and QTP inland river basins (also known as the Qiangtang Plateau), with the POD generally being <0.60.

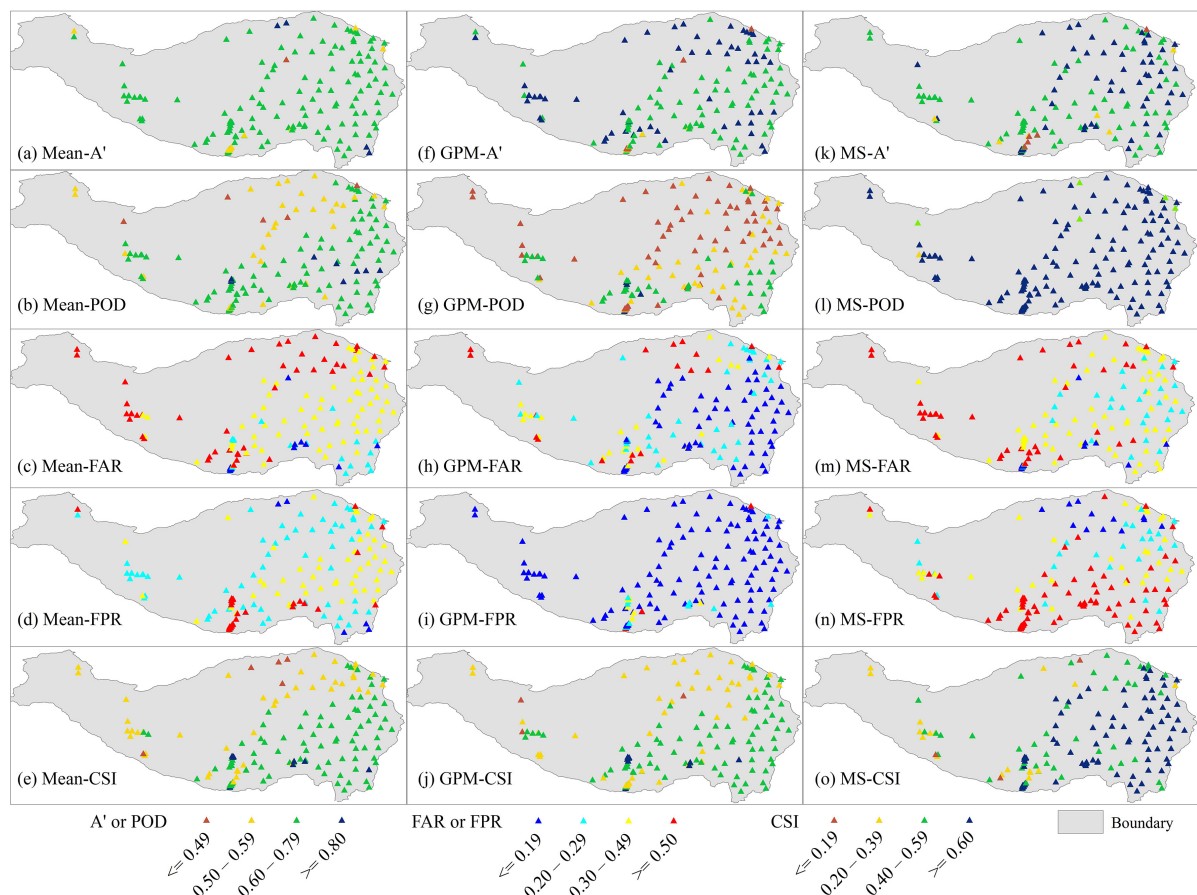

**Figure 4.** Spatial performance of SPPs at detecting precipitation occurrence in the QTP, including the MSWEP and GPM showing better performance than the other products, and the average performance of all products.

The evaluation of the CSI showed a similar spatial distribution. The performance of the SPPs in the east and south were substantially better than those in the west (chiefly

including the source regions of Tarim and Indus river basins) and north (chiefly including the QTP inland river basins). CSI values were generally >0.40 at the eastern and southern stations, and generally <0.40 at the northern and western stations.

The performance of satellite products evaluated by FAR exhibited an increasing trend from the southeast to northwest. FAR was generally <0.30 at the southern edge and in the southeast (Hengduan Mountains), and that of some stations was even <0.20. FAR was generally >0.50 in the western, northern, and southern regions. Except for the Hengduan Mountains in the southeast, the spatial performance of FPR evaluation was consistent with that of FAR, that is, FPR was generally lower (<0.30) in the east and south, and higher in the west and north (>0.50). Contrary to the performance of the FAR, the FPR of SPPs was generally >0.50 in the Hengduan Mountains.

When performing the evaluation using the A' index, the satellite products had a strong performance at detecting precipitation occurrence. The A' of most stations (151/161) was >0.60, with values >0.70 at 65% (105/161) of the stations. The stations with A' < 0.60 were chiefly located in the northwestern, northeastern, and southern edges of the QTP.

The spatial performance of GPM was consistent with the average performance of all SPPs in the evaluation of A' and CSI. GPM performed better than average across all products, with A' > 0.70 at 80% of the stations (128/161). GPM showed strong capture ability, with POD > 0.60 only at the southern stations. POD was generally <0.50 at the western and north-central stations, indicating that the GPM showed poor performance at correctly detecting precipitation occurrence in these regions. However, GPM performed substantially better than average in the FAR and FPR.

MSWEP showed a strong ability to capture the occurrence of precipitation. The POD at most stations (155/161) was >0.80. The number of stations (91/161) with CSI values >0.60 was also much higher than for the other products, especially in the eastern QTP. However, MSWEP also showed a high false alarm rate/ratio. The spatial distribution of MSWEP in the evaluation of A' was consistent with that of GPM.

The performance of SPPs was also analyzed in different periods: in the warm (June–September) and cold (December–March) seasons, and on an annual scale (Figure 5). The ability of SPPs to capture precipitation occurrence varied substantially in different periods, with the mean POD varying between 0.38 and 0.96. Overall, the capture ability of SPPs in the warm season was substantially higher than that in the cold season. The mean POD of all stations for each product was mainly concentrated between 0.58 and 0.80 (25–75% frequency range) in the warm season. With the exception of CHIRPS and PERSIANN-CCS, all other products showed strong ability, with mean POD > 0.60. The mean POD was concentrated between 0.29 and 0.68 in the cold season. With the exception of MSWEP, PERSIANN, PERSIANN-CCS, and PERSIANN-CDR, all other SPPs showed poor capture ability in the cold season, with the mean POD being <0.50 for most products. The capture ability of SPPs generally showed a unimodal distribution throughout the year. The ability was weakest in December, and strongest in July and August.

The performance evaluation of FAR also varied substantially in different periods. Overall, FAR in the cold season was lower than in the warm season. The mean FAR of each product was between 0.13 and 0.72 in the warm season. CHIRPS showed the strongest performance, with the lowest FAR. The mean FAR of each product in the cold season ranged from 0.03 to 0.44. GPM showed the strongest performance, with the lowest FAR. FAR had the strongest performance in December, with the mean values of FAR being between 0.02 and 0.30. GPM showed the strongest performance. CHIRPS, GSMaP-MVK, and GSMaP-NRT also showed better performance than the other SPPs.

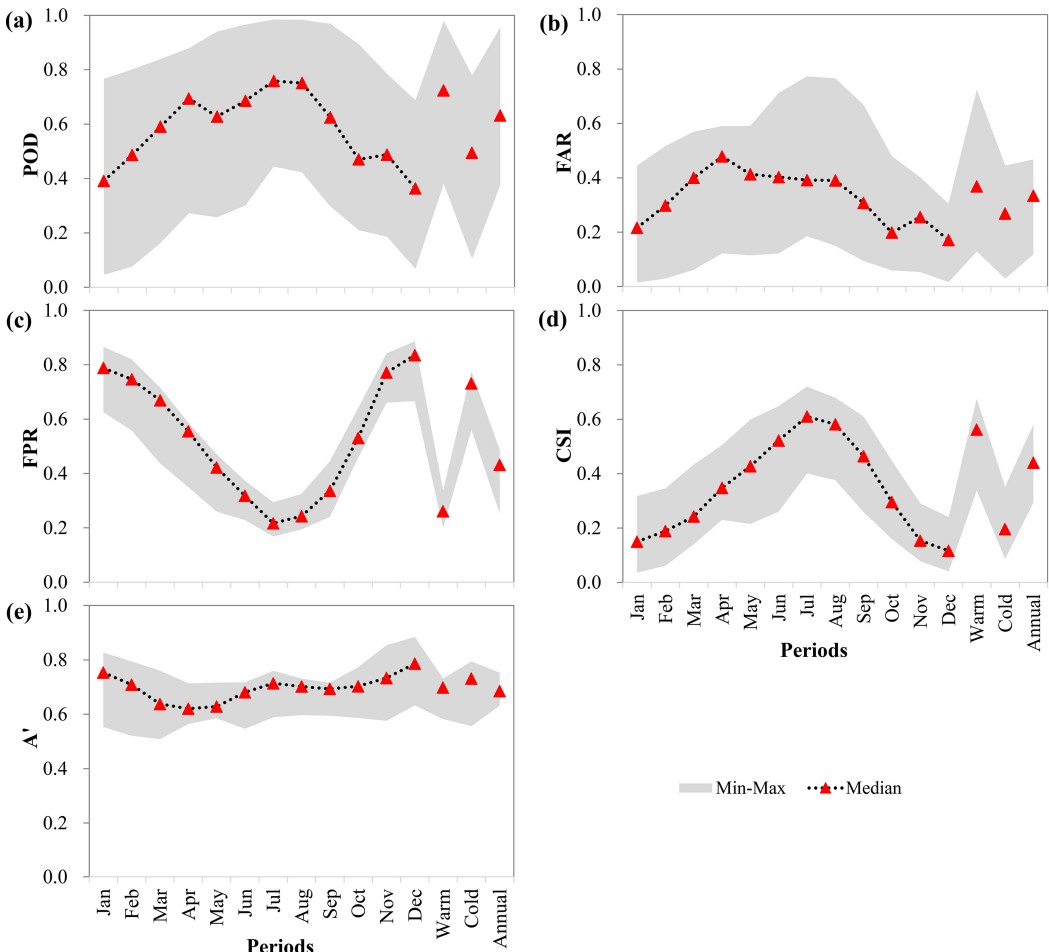

**Figure 5.** Performance of SPPs at detecting precipitation occurrence during different periods ((**a**–**e**) are the performance of SPPs evaluated by the POD, FAR, FPR, CSI, and A′, respectively. Min and Max are minimum and maximum values, respectively).

The performance of SPPs evaluated using the FPR exhibited little difference between periods. The standard deviation calculated from the mean FPR of each SPP was approximately 0.06 in each period. This indicated that the detection ability of the SPPs was relatively consistent among the satellite products. This is in contrast to the results evaluated using POD and FAR. FRP was much higher in the cold season than in the warm season. The mean FPR values of each SPP were concentrated in the range 0.69–0.75 in the cold season. In the warm season, they ranged from 0.20 to 0.34. Monthly FPR showed a regular single-valley distribution throughout the year. The strongest performances were found in CHIRPS and GPM.

The performance evaluation of CSI also showed little difference, with the standard deviation being approximately 0.08 in each period. The SPPs showed the strongest ability in July, with mean CSI values between 0.54 and 0.65. Performance was poor in the months of the cold season, with the mean CSI < 0.45. The performance in the warm season was also substantially better than that in the cold season. The mean CSI of each SPP was close to or >0.50, except for CHIRPS.

The evaluation of the performance of A′ also showed little difference, with a standard deviation of approximately 0.06 in most periods, except during January and February. Performance evaluated using the POD, FPR, FAR, and CSI showed a seasonal pattern during the year. However, performance evaluated by A′ did not show seasonal variations. The performance of SPPs from March–May with the mean A′ < 0.65, was slightly lower than that of the other months.

Overall, the spatial and temporal performance of SPPs in detecting precipitation occurrence showed variations when evaluated by different indices. In general, the SPPs tend to exhibit better performance in east and south than that in west and north, when evaluated using the POD, FAR and CSI indices. Performance in the warm season was also substantially better than that in the cold season when evaluated using the POD, FPR and CSI indices. However, the spatial and temporal performance of SPPs showed little difference in the A′ index.

### 3.3. Performance of SPPs at Measuring Precipitation Series

Figure 6 shows the ability of satellite products to estimate monthly precipitation. RE generally ranged from −80% to 80% at the stations. The mean RE of each SPP was between −66% and 46%. Most SPPs (7/10) showed an overestimation in the QTP. The mean overestimation of each SPP ranged from 19 to 46%, with a mean value of 27%. GPM, which had the strongest ability in the RE, underestimated QTP precipitation by 13%. However, it varied substantially among stations, with only 27% of stations (43/161) having an RE within ±20%. PDIR-Now, CMORPH, and CHIRPS also showed better performance, with an overestimation of approximately 20%. PERSIANN and PERSIANN-CCS had the weakest performance at estimating the precipitation amount, with an underestimation of approximately 65%. SPP measurements generally showed positive correlation with observed precipitation in the QTP, except for the PERSIANN and PERSIANN-CCS. MSWEP had the strongest correlation, with the mean CC at 0.87. CC was >0.70 at 94% (152/161) of the stations. The PERSIANN-CDR and CHIRPS also showed a strong correlation, with 86% of stations having CC > 0.70. The mean CC of the two products were 0.82 and 0.81, respectively.

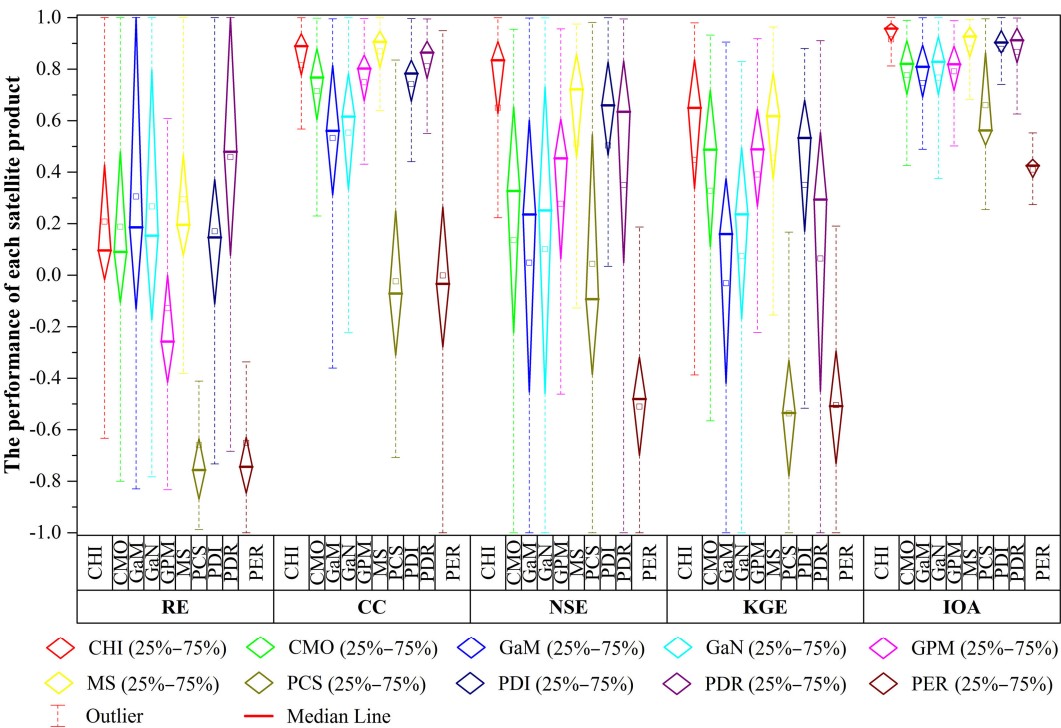

**Figure 6.** Performance of SPPs at estimating monthly precipitation.

The CHIRPS measurements were in good agreement with the observed precipitation in the QTP, with a mean NSE at 0.65. Measurements of the PDIR-Now and MSWEP also matched well with the observed values, with the mean NSE values of the two products being 0.51 and 0.50, respectively. However, the other products showed relatively poor agreement with monthly precipitation, with the mean NSE being <0.35 in each

product. The SPPs showed poor performance in the evaluation of KGE, with the mean value of KGE being only 0.40. MS and CHIRPS showed the strongest performance, with KGE > 0.50 at approximately 63% of the stations. The variation in the products among stations evaluated using IOA was substantially smaller than that evaluated using the other four indices. CHIRPS showed the strongest performance, with IOA > 0.80 at 89% of the stations. MSWEP, PERSIANN-CDR, and PDIR-Now also showed better performance, with the IOA > 0.80 at 86%, 87%, and 80% of the stations, respectively.

The spatial performance of SPPs at estimating monthly precipitation is shown in Figure 7. The RE of CHIRPS was within ±20% at most stations (88/161). However, great overestimation (RE > 40%) was observed at 42 stations, mainly located in the southwestern and northeastern regions. MSWEP overestimated precipitation at most stations (142/161). The overestimation was higher in the northern and southwestern stations. From the mean performance of all SPPs, RE was within ±20% at 61% of the stations, chiefly located in the eastern regions. There were 30 stations with the RE > ±40%, chiefly located in the western and northern regions. The overestimation was <20% at the southeastern stations, while it was >40% at the western stations.

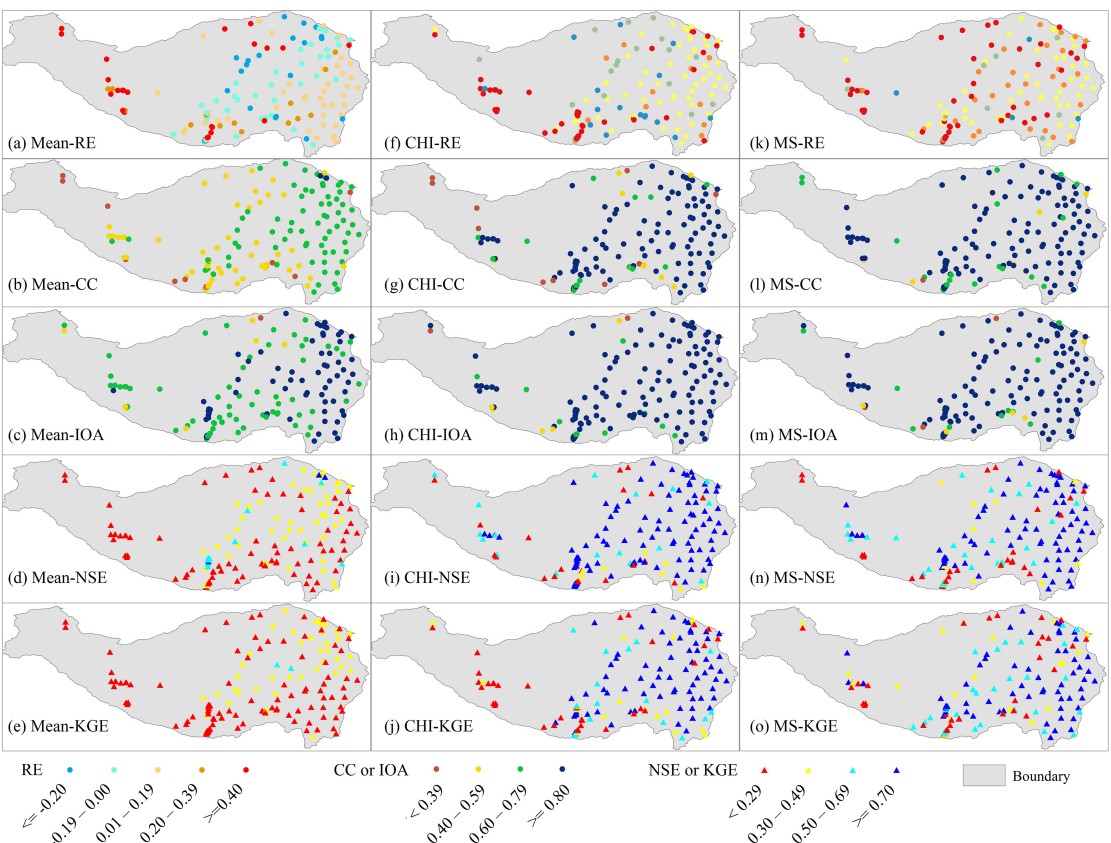

**Figure 7.** Spatial performance of SPPs at estimating monthly precipitation in the QTP, including CHIRPS and MSWEP showing better overall performance than the other products, and the average performance of all SPPs.

The other four indices showed that the mean performance of all products was much lower than that of CHIRPS and MSWEP. Only a few stations had CC > 0.80 for the mean performance of all products. CHIRPS and MSWEP had CC > 0.80 at 119 and 131 stations, respectively. The CC was substantially higher at the eastern stations than at the western stations. The results of the IOA were similar to those of CC. For the mean performance of all SPPs, the NSE and KGE values were <0.50 at 87% and 99% of stations, respectively. CHIRPS and MSWEP performed well at most stations, with NSE > 0.70 at 70% and 52% of stations, respectively. These stations were mainly located in the eastern QTP. There were

also several stations with poor performance, chiefly located in the western regions, and in the margins of the northern and southern regions.

Overall, except RE, the evaluation of all of the other four series indices showed that CHIRPS and MSWEP outperformed the other products in measuring monthly precipitation in the QTP. Only a few SPPs were able to capture the precipitation series, with the mean NSE of all stations being >0.50, including CHIRPS, MSWEP and PDIR-Now. The performance of CHIRPS and MSWEP was generally better in the eastern QTP than in the western regions.

### 3.4. Comprehensive Evaluation of SPPs

The overall performance of satellite products was quantitatively evaluated using the improved RS method on the basis of 10 indices (Table 2). The evaluation of the five occurrence indices showed that the strongest products were concentrated in the MSWEP and GPM. MSWEP showed the strongest performance in the CSI and POD indices, indicating that it had the strongest ability to capture precipitation occurrence in the QTP. GPM showed the strongest performance in the FAR, FPR, and A′ indices, indicating that it had the strongest comprehensive capture capability. CHIRPS showed the weakest ability to detect precipitation occurrence, with the lowest values of the POD and CSI. PERSIANN showed the weakest performance in the FAR and A′ indices. This was confirmed through a comprehensive assessment of five precipitation occurrence indices using the improved RS method. GPM had the highest RS value, at 0.75. The comprehensive performance of the GSMaP-NRT and MSWEP on precipitation occurrence was better than the other SPPs, with RS values of 0.62 and 0.65, respectively. CHIRPS and PERSIANN performed relatively poorly, with RS values of 0.22 and 0.21, respectively.

**Table 2.** The overall performance of each SPP evaluated on the basis of each index.

| SPPs | A′ | CSI | FAR | FPR | POD | CC | IOA | KGE | NSE | RE | RSM | RSO | RSC |
|------|-----|-----|------|------|------|------|------|-------|-------|-------|------|------|------|
| CHI | 0.67 | <u>0.29</u> | 0.39 | 0.14 | <u>0.38</u> | 0.82 | **0.92** | 0.45 | **0.65** | 0.21 | **0.92** | 0.33 | 0.62 |
| CMO | 0.67 | 0.44 | 0.45 | 0.38 | 0.70 | 0.71 | 0.78 | 0.33 | 0.14 | 0.19 | 0.63 | 0.36 | 0.50 |
| GaM | 0.74 | 0.44 | 0.43 | 0.26 | 0.64 | 0.53 | 0.75 | −0.03 | 0.05 | 0.31 | 0.39 | 0.54 | 0.46 |
| GaN | 0.74 | 0.46 | 0.36 | 0.23 | 0.62 | 0.55 | 0.77 | 0.07 | 0.10 | 0.27 | 0.45 | 0.62 | 0.54 |
| GPM | **0.75** | 0.44 | **0.25** | **0.12** | 0.53 | 0.75 | 0.79 | 0.39 | 0.28 | **−0.13** | 0.74 | **0.75** | **0.75** |
| MS | 0.74 | **0.58** | 0.40 | <u>0.47</u> | **0.96** | **0.87** | 0.88 | **0.46** | 0.51 | 0.29 | 0.84 | 0.65 | **0.75** |
| PCS | <u>0.63</u> | 0.37 | 0.47 | 0.33 | 0.60 | <u>−0.02</u> | 0.66 | <u>−0.54</u> | 0.04 | <u>−0.66</u> | <u>0.30</u> | 0.22 | <u>0.26</u> |
| PDI | 0.70 | 0.48 | 0.43 | 0.37 | 0.79 | 0.74 | 0.88 | 0.35 | 0.50 | 0.17 | 0.82 | 0.49 | 0.65 |
| PDR | 0.66 | 0.46 | 0.47 | 0.44 | 0.79 | 0.81 | 0.87 | 0.06 | 0.35 | 0.46 | 0.56 | 0.33 | 0.44 |
| PER | <u>0.63</u> | 0.37 | <u>0.49</u> | 0.34 | 0.61 | 0.00 | <u>0.41</u> | −0.50 | <u>−0.51</u> | −0.65 | 0.34 | <u>0.21</u> | 0.28 |

Note: RSC is the RS value evaluated on the basis of all ten indices. RSM and RSO are RS values evaluated on the basis of series and occurrence indices, respectively. A′ is the index of Accuracy. Underlined values indicate the worst performance, while bold values indicate the best performance.

The evaluation of five series indices showed that the SPPs with the strongest performance were concentrated in CHIRPS, GPM and MSWEP. CHIRPS had the strongest performance in the IOA and NSE indices, indicating that the CHIRPS measurements were in good agreement with the observed series. MSWEP had the strongest performance in the CC and KGE indices. GPM measurements showed that it possessed the lowest bias with respect to the mean precipitation over the QTP. The comprehensive evaluation of series indices showed that CHIRPS had the strongest performance, with the RS value reaching 0.92. PDIR-Now and MSWEP also showed better performance than the other SPPs, with RS values of 0.84 and 0.82, respectively. PERSIANN-CCS showed the weakest comprehensive performance, with an RS value of 0.30.

In the comprehensive evaluation of all ten indices, GPM and MSWEP outperformed the other SPPs, with an RS value of 0.75. PDIR-Now and CHIRPS also showed better overall performance than the other SPPs, with RS values >0.60. The PERSIANN and PERSIANN-CCS showed the weakest comprehensive performance, with RS values of 0.28 and 0.26, respectively.

## 4. Discussions

### 4.1. Limitations, Uncertainties and Novelty of the Study

The novelty of this study lies mainly in its quantitative comprehensive evaluation. The comprehensive evaluation results previously obtained using SPPs were typically subjectively affected. In other words, it is easy to obtain different comprehensive evaluation results. In this study, based on the improved RS method, the comprehensive evaluation of SPPs was quantitatively evaluated using multiple indices. The improved RS method provided comprehensive and objective evaluation results.

The new ground observation network, which was set up by researchers, was included in this study to increase the reliability of the results in this data-scarce region. There were more SPPs evaluated in this study than in previous studies. Several new SPPs were found to perform better than the satellite products used for evaluation in previous studies. For example, the PDIR-Now, PERSIANN-CDR, and MSWEP, which had not been included in previous evaluation studies for the QTP, had the strongest performance in terms of POD. The mean POD values of these products were 0.96, 0.79, and 0.79, which is higher than the values of the optimal SPPs (<0.75) previously evaluated [10,28–30].

The UTC of the SPPs was inconsistent with the Beijing time of the ground observation. The conversion of time is described in Section 2.1.1. There were also several NTPDC stations that provided only daily data. The SPPs with hourly precipitation were converted to Beijing time at these stations. However, there were still a few SPPs providing only daily series. This time inconsistency could have affected the evaluation results. A comparison of the evaluation results showed that the time inconsistency only had a slight effect on each occurrence index. It did not change the comparative evaluation results among the different SPPs. The time inconsistency also had a small effect on the evaluation results of series indices.

The lengths of the observed precipitation series were different between the CMA and NTPDC stations. At present, the series acquired by many NTPDC stations are relatively short, with the length of observation being <1 year at some stations. The author will attempt to obtain more extensive and longer observation data for future studies.

### 4.2. The Performance of SPPs with Respect to Precipitation Levels, Indices, Spatial and Temporal Distribution

SPPs showed the strongest ability to detect the occurrence of precipitation at levels of 10–25 mm, when evaluated using the FAR and CSI indices. The performance was relatively poor at capturing rainstorm at the QTP. This was consistent with evaluation results reported by Lei et al. [30] in the eastern QTP. The performance of SPPs was generally positively correlated with precipitation levels when evaluated using the A′, POD, and FPR indices. The higher the precipitation amount, the better the performance of SPPs.

The performance of SPPs varied substantially in different evaluation indices. Different indices might give opposite evaluation results. For example, SPPs showed better performance in the cold season than in the warm season when evaluated using the A′ and FAR occurrence indices. However, the evaluation results of the CSI, POD, and FPR indices of the same type were the opposite (Figure 5). CHIRPS showed the weakest performance with POD and CSI among the occurrence indices, while it had the strongest performance among the series indices for NSE and IOA (Table 2). Therefore, multiple indices are recommended for the evaluation of SPPs in order to obtain comprehensive results. This might also be useful for the evaluation of other products or models.

In general, the SPPs showed better performance in the southern and eastern QTP. These regions are mainly located in the summer monsoon area, with relatively dense precipitation stations. The performance was poor in the northern and western QTP. These regions are in the westerly circulation area with few on-ground observation stations. The relatively dense stations provided more reliable ground fusion data input for satellite precipitation measurements. Conversely, precipitation under the influence of the summer monsoon has strong seasonal regularity. However, the seasonal precipitation in the westerly-

circulation-affected area is weak. The SPPs tended to show better performance at measuring precipitation with strong seasonality. This was consistent with the simulation accuracy of the evaluation results of actual evapotranspiration and surface evaporation on the global scale [97,98]. The limited number of stations in the western part of the QTP might affect the reliability of the evaluation results in this region.

GPM has previously been reported as having the strongest performance [10,30]. This study also found GPM to show the strongest overall performance in the QTP. In addition, it had the strongest performance at capturing false alarm rate/ratio and mean values. Although most previous evaluations showed that the GPM had the strongest performance, this was mainly because these evaluations focused on the indices of precipitation occurrence. This study also found that it showed the strongest overall performance with regard to precipitation occurrence indices. However, the performance of GPM in the evaluation of series indices was lower than that of CHIRPS, MSWEP, and PDIR-Now. Lei et al. [30] found that CHIRPS showed the weakest performance in the QTP based on the evaluation of precipitation occurrence indices. However, the overall performance of CHIRPS was the strongest among the 10 SPPs when evaluated using the series indices in this study.

Although the SPPs showed the ability to detect precipitation occurrence and series in the QTP, their performance in the QTP was relatively poor compared with that in other areas in China. For example, Wei et al. [24] found that the KGE of the optimal SPPs in eastern China was >0.90, with values >0.70 for most SPPs. In this study, comprehensive evaluation compared to both the CMA and NTPDC stations showed the mean KGE was <0.50 for each of the SPPs. Tang et al. [22] found the mean CSI of the optimal SPPs in China to be close to 0.70. However, in this study, the mean value of CSI for the optimal SPPs in the QTP was only 0.58, which is much lower than that in China overall. Overall, the accuracy of satellite products in the QTP needs further improvement. The QTP is a relatively active area for convective clouds and mesoscale convective systems [99]. Therefore, precipitation processes on the QTP are complex, especially with respect to convective cloud precipitation [100]. This increases the difficulty of performing satellite precipitation estimates. It should be noted that in addition to the variations in accuracy performance, the SPPs evaluated in this study also have their own advantages. For example, PERSIANN CDR aims to provide long-term continuous precipitation estimates, so it is expected that it will have lower accuracy than GPM. Therefore, it is suggested that these issues could be considered in the application of SPPs.

## 5. Conclusions

Ten high-resolution SPPs were quantitatively and comprehensively evaluated from ten indices using an improved RS method in the QTP. Overall, the GPM and MSWEP showed the strongest performance, with RS values of 0.75. CHIRPS and PDIR-Now also showed better overall performance than the other SPPs, with RS values >0.60. The PERSIANN and PERSIANN-CCS showed the weakest comprehensive performance. CHIRPS and GPM had the strongest performance for measuring precipitation occurrence (RS = 0.92) and series (RS = 0.75), respectively. While the optimal SPPs varied in terms of the indices used, they were concentrated in the MSWEP, GPM and CHIRPS. MSWEP showed the strongest performance in the POD, CSI, CC, and KGE indices, indicating that it had the highest ability to detect precipitation occurrence, and the strongest correlation with observed values in the QTP. GPM showed the strongest performance in the FAR, FPR, A′, and RE indices, indicating that GPM had the lowest bias and false alarm ratio/rate, and the strongest comprehensive capture capability.

Performance of SPPs varied substantially, in both products and stations. The SPPs tended to overestimate precipitation in the QTP, with the mean overestimation of seven SPPs being 27%. The bias of SPPs was marked in the QTP, with the RE generally being between $-80\%$ and 80%. Even GPM, which showed the strongest performance in terms of RE, only had RE within $\pm20\%$ at 27% of stations (43/161). In general, most of the SPPs showed the ability to detect precipitation occurrence. The mean values of A′, CSI, FAR, FPR,

and POD for all SPPs across all stations were 0.69, 0.43, 0.41, 0.31, and 0.66, respectively. SPPs showed relatively weak performance at measuring precipitation series. Only a few SPPs could capture precipitation series, with the mean NSE of all stations being >0.50, including CHIRPS, MSWEP and PDIR-Now. The SPP measurements over the QTP need to be further improved, especially with respect to the measurement of precipitation series.

The performance of SPPs was generally positively correlated with the precipitation levels evaluated using the POD, FPR and A′ indices. The higher the amount of precipitation, the better the performance of SPPs. The SPPs showed the highest ability to capture precipitation occurrence at the level of 10–25 mm when evaluated using the CSI and FAR indices. SPPs showed better performance in the southern and eastern QTP, chiefly including the Yangtze, Yellow, Nu–Salween, Lancang–Mekong, Yarlung Zangbo–Bramaputra, and Ganges river basins. These basins are mainly affected by the East Asian monsoon and South Asian monsoon, with relatively dense precipitation stations. The performance was relatively poor in the western and northern QTP, chiefly including the Tarim, Indus, and QTP inland river basins. These basins are mainly located in the westerly circulation area, and there are few on-ground observation stations.

In general, the performance of SPPs showed a seasonal pattern during the year in all precipitation occurrence indices except for A′ index. The evaluation results were consistent in each period when evaluated using the POD, CSI and FPR indices. The strongest and weakest performances were in July and December, respectively. SPPs showed better performance in the warm season than in the cold season. However, the evaluation results of the FAR and A′ indices were opposite, that is, the performance in the cold season was slightly better than that in the warm season. Therefore, multiple indices are recommended for the evaluation of SPPs in order to obtain comprehensive evaluation results.

**Supplementary Materials:** The following supporting information can be downloaded at: https://www.mdpi.com/article/10.3390/rs15133381/s1. Table S1: Observation stations of National Tibetan Plateau Data Center used in this study.

**Funding:** This study was supported and funded by the Second Tibetan Plateau Scientific Expedition and Research Program (2019QZKK1006) and the National Natural Science Foundation of China (42171029).

**Data Availability Statement:** The codes, materials, and data can be obtained by contacting at zfliu@igsnrr.ac.cn.

**Acknowledgments:** The author is grateful to the public data portal for providing the observed and satellite rainfall products datasets. The observation dataset is provided by China Meteorological Data Service Center (http://data.cma.cn/ (accessed on 28 January 2023)) and National Tibetan Plateau Data Center (http://data.tpdc.ac.cn (accessed on 28 January 2023)).

**Conflicts of Interest:** The author declares no conflict of interest.

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
