# Peer review of "Comprehensive Evaluation of High-Resolution Satellite Precipitation Products over the Qinghai–Tibetan Plateau Using the New Ground Observation Network"

_remotesensing, doi:10.3390/rs15133381_

Round 1

Reviewer 1 Report (New Reviewer)

This paper provides a comprehensive evaluation on 10 kinds of satellite precipitation products (SPPs) over the Qinghai-Tibetan Plateau (QTP) region. An improved rank score method and multiple indicators were used based on the data from the China Meteorological Administration (CMA) and the National Tibetan Plateau Data Centre (NTPDC). Overall the works are massive and substantial, but innovation is somewhat lacked since the evaluation methods and indicators used in this study are actually common ones.

Some comments and suggestions are given as follow.

1. Multiple high-resolution satellite precipitation products were compared, but what is the reason for choosing these specific products instead of others? These products have different data sources, processing methods, and target objectives, naturally resulting in different accuracy. Is it appropriate to compare these products together? For example, PERSIANN CDR aims to provide long-term continuous precipitation observations, so it is expected to have lower precipitation detection and estimation accuracy compared to GPM.

2. What are innovative aspects of this study? The conclusions seem to largely agree with previous studies, and there is no impressive findings from this study. The "improved rank score method" was stressed by the author but it seems that such a method does not play a key role in this study.

3. L23. What does "KGE" mean? The abbreviation is not mentioned in the text.

4. L72-L73. What does "other products" refer to? The relevant statements are not very appropriate, especially near "the GPM showed strongest performance."

5. L114-L115. It is mentioned that the site dataset is provided by NTPDC, but the paper does not provide detailed information about the time range of the data from these stations. Additionally, the detailed information regarding all data used in the study, such as the time range and data volume, is not explained.

6. L189. The "RS method" lacks necessary descriptions. What are its characteristics and purposes? The author mentions it as an "improved one", but what are its evident superiorities?

7. L227-L281. Here and subsequent sections that discuss spatial performance, satellite precipitation products often exhibit poorer performance in the western part of the Tibetan Plateau. Is this conclusion partly due to the spatial distribution density of ground stations? In the eastern part of the Tibetan Plateau, where stations are denser, are the results more reliable? Does the limited number of stations in the western part of the Tibetan Plateau affect the reliability of the results? This needs to be carefully analyzed and clarified.

8. L324-L329. It is mentioned how various satellite precipitation products perform in different month, but this aspect is not reflected in Figure 5.

9. L351. Why did the author choose to compare the monthly precipitation performance of multiple satellite precipitation products? Given the coarsest time resolution of daily that is common for all these precipitation data, why the comparison and analysis was not on the time scale of daily?

10. What are the main conclusions of Sections 3.1, 3.2, and 3.3? Could the author provide a concise and clear summary? These sections describe the performance of various satellite precipitation products on different evaluation indicators in detail, but the descriptions appear fragmented, in turn not very clear.

11. L441-L442. The statement "Previous comprehensive evaluation results of SPPs were usually subjectively affected" seems somewhat subjective. Does the author have sufficient evidence to support this argument?

12. L485-L466. The author mentions that "This study tried to compare the evaluation results between the long and short series stations and did not find any obvious differences". Evidences should be provided.

Many words are inappropriate. For example, L25, the westerly areas.

Author Response

Responses to Reviewer’ Comments
Manuscript ID: remotesensing-2448871
Title: Comprehensive evaluation of high-resolution satellite precipitation products over the Qinghai-Tibetan Plateau using the new ground observation network
Author(s): Zhaofei Liu
It would be greatly appreciated for your kind reviewing to this paper. The manuscript has been thoroughly revised according to your valuable comments and suggestion. The revised manuscript should, after incorporating your valuable advices and suggestions, be improved greatly. For your convenience to re-review the paper, the response corresponding to your comments are described in detail as follows:
Comments and Suggestions for Authors
This paper provides a comprehensive evaluation on 10 kinds of satellite precipitation products (SPPs) over the Qinghai-Tibetan Plateau (QTP) region. An improved rank score method and multiple indicators were used based on the data from the China Meteorological Administration (CMA) and the National Tibetan Plateau Data Centre (NTPDC). Overall the works are massive and substantial, but innovation is somewhat lacked since the evaluation methods and indicators used in this study are actually common ones.
Some comments and suggestions are given as follow.
1. Multiple high-resolution satellite precipitation products were compared, but what is the reason for choosing these specific products instead of others? These products have different data sources, processing methods, and target objectives, naturally resulting in different accuracy. Is it appropriate to compare these products together? For example, PERSIANN CDR aims to provide long-term continuous precipitation observations, so it is expected to have lower precipitation detection and estimation accuracy compared to GPM. 
Reply: The author selects these products mainly based on the availability of these products. Yes, these products are different in data sources, method, and objectives. According to your valuable comments, in Lines 566-570, several new sentences “It should be noted that in addition to the variations in accuracy performance, the SPPs evaluated in this study also have their own advantages. For example, PERSIANN CDR aims to provide long-term continuous precipitation estimates, so it is expected to have lower accuracy compared to GPM. Therefore, it is suggested that these issues could be considered in the application of SPPs.” have been added to discuss this issue. 

2. What are innovative aspects of this study? The conclusions seem to largely agree with previous studies, and there is no impressive findings from this study. The "improved rank score method" was stressed by the author but it seems that such a method does not play a key role in this study.
Reply: Thanks for your comments. As mentioned in Lines 11-12, “Satellite precipitation products (SPPs) have been widely evaluated at regional scales. However, there was little quantitative comprehensive evaluations of SPPs using multiple indices.” There are many studies quantitatively evaluated SPPs by each index. However, the comprehensive evaluation results were generally obtained qualitatively. For example, the model 1 might showed the best performance in the A, B, and C evaluation indices, while the model 2 was the best in D and E indices. In these evaluation based on each index, comprehensive evaluation results might be subjectively made that the best model was the model 1 or model 2. In this study, an improved rank score (RS) method was developed for the quantitative comprehensive evaluation of SPPs. The proposed improved RS method can also potentially be applied for comprehensive evaluation of other products and models. In addition, as for the study area, both the new observation network and national basic stations were used to obtain reliable comprehensive evaluation results for the data scarce Qinghai-Tibetan Plateau.
Improvements of this study are discussed in the Section “4.1 Limitations, uncertainties and improvements of the study”. Firstly, there are some consistent results with previous studies, but there are also some new and different results. Please find details in this section. Secondly, there are several evaluations of SPPs over the study area (QTP). However, previous studies have only used national stations of the China meteorological administration (CMA), sparsely distributed in the QTP. The national Tibetan plateau data center (NTPDC) developed a new precipitation observation network comparable to the number of CMA stations. This provides a new and wider ground observation network for evaluating SPPs in the QTP. At present, SPPs evaluations based on the new observation network are lacking. The new observation network comparable with the number of national basic stations were applied for the SPPs evaluation to obtain more reliable results compared with previous studies, which only used basic stations. Finally, this study focus on Comprehensive evaluation of high-resolution satellite precipitation products over the Qinghai-Tibetan Plateau using the new ground observation network. Comprehensive evaluation based on the improved rank score method is an important content in this study. 

3. L23. What does "KGE" mean? The abbreviation is not mentioned in the text.
Reply: Thanks very much for your valuable comments. In Line 23, the word “KGE” has been revised by “Kling-Gupta efficiency”.

4. L72-L73. What does "other products" refer to? The relevant statements are not very appropriate, especially near "the GPM showed strongest performance."
Reply: Thanks very much for your valuable comments. In Lines 74-76 (the revised manuscript), the word “other products” has been revised by “precipitation estimation from remotely sensed information using artificial neural net-works (PERSIANN) and tropical rainfall measuring mission (TRMM) products”.

5. L114-L115. It is mentioned that the site dataset is provided by NTPDC, but the paper does not provide detailed information about the time range of the data from these stations. Additionally, the detailed information regarding all data used in the study, such as the time range and data volume, is not explained.
Reply: Thanks very much for your valuable comments. The detailed information about the site dataset is shown in Table S1 in the Supplementary Materials. According to your valuable comments, the time range and temporal resolution have been added in the revised Supplementary Materials. Please find details in the revised Table S1.

6. L189. The "RS method" lacks necessary descriptions. What are its characteristics and purposes? The author mentions it as an "improved one", but what are its evident superiorities?
Reply: The sections 2.2.1 and 2.2.2 are evaluation of SPPs based on each index. As mentioned in the first sentence of the section 2.2.3, the main characteristics and purposes of the RS method is “comprehensively evaluates performance of the model by using multiple indices”. It subjectively merges RS values into integers ranging from 0–9. The improved RS method could “objectively evaluated the comprehensive performance of a model by using the deviation degree between each model and the optimal model value in each index.” (Lines 197-198). 
According to your valuable comments, the new sentences have been added in the revised manuscript to make it more clear, as follows,
In Line 192, “The above is evaluation of SPPs based on each index.”
In Lines 195-196, “and subjectively merges RS values into integers ranging from 0–9.”

7. L227-L281. Here and subsequent sections that discuss spatial performance, satellite precipitation products often exhibit poorer performance in the western part of the Tibetan Plateau. Is this conclusion partly due to the spatial distribution density of ground stations? In the eastern part of the Tibetan Plateau, where stations are denser, are the results more reliable? Does the limited number of stations in the western part of the Tibetan Plateau affect the reliability of the results? This needs to be carefully analyzed and clarified.
Reply: These conclusions are mainly derived from the proportion of sites with better or poorer performance. Take the Figure 4-e as an example, CSI was <0.40 at 85% of stations (11/13) in the western part of the QTP. It was >0.40 at more than 90% of stations in the southeastern region. Therefore, it is concluded that SPPs exhibit poorer performance in the western part of the QTP, while showed better performance in the southeastern region. However, yes, the limited number of station in the western region might affect the reliability of the results. According to your valuable comments, in Lines 542-543, a new sentence “The limited number of stations in the western part of the QTP might affect the reliability of the evaluation results in this region.” has been added to discuss this issue. 

8. L324-L329. It is mentioned how various satellite precipitation products perform in different month, but this aspect is not reflected in Figure 5.
Reply: Figure 5 showed performance of SPPs in detecting precipitation occurrence during different periods. The horizontal coordinates in the figure are different periods, including months, warm and cold seasons. The ordinate gives a range of values for each index of SPPs in each period. In the figure, “Min” and “Max” are minimum and maximum values in the value range. 

9. L351. Why did the author choose to compare the monthly precipitation performance of multiple satellite precipitation products? Given the coarsest time resolution of daily that is common for all these precipitation data, why the comparison and analysis was not on the time scale of daily?
Reply: Evaluation of SPPs by precipitation occurrence indices was based on the time scale of daily. However, as for time series indices, there are many zeros in the daily precipitation series at the study area, because many stations are located in arid and semi-arid regions. Although there is no difference in the calculation results of relative error between daily and monthly series, they have great effects on the calculation results of the other four indices. The variation of these indices calculated from daily series is much smaller than that calculated from monthly series. In other words, the application of monthly series for evaluation better reflects the differences among products and stations. Take the index of Pearson’s correlation coefficient (CC) as an example, the CC values of stations calculated from daily series are generally between 0.05 and 0.15, while those for monthly series are mainly ranged from 0.20 to 0.95. 

10. What are the main conclusions of Sections 3.1, 3.2, and 3.3? Could the author provide a concise and clear summary? These sections describe the performance of various satellite precipitation products on different evaluation indicators in detail, but the descriptions appear fragmented, in turn not very clear.
Reply: Three new paragraphs have been added in the revised manuscript to give main conclusions of Sections 3.1, 3.2, and 3.3, respectively. They are as flows, 
Main conclusions of Sections 3.1 are added in Lines 292-299, “Overall, SPPs showed the ability to detect a fraction of precipitation occurrences and were with many false alarms in the QTP. The MSWEP and GPM generally showed the strongest performance when evaluated by precipitation occurrence indices. In general, SPPs performance showed a positive correlation with precipitation levels in the evaluation of the POD, FPR and A’ indices. With the increase of precipitation amount in the annual cycle, the performance of SPPs first increased and then decreased when evaluated by the CSI and FAR indices. Satellite products had the strongest performance in capturing precipitation between 10–25 mm.”
Main conclusions of Sections 3.2 are added in Lines 386-392, “Overall, the spatial and temporal performance of SPPs in detecting precipitation occurrence showed variations when evaluated by different indices. In general, the SPPs tend to exhibit better performance in east and south than that in west and north, when evaluated by the POD, FAR and CSI indices. Performance in the warm season was also substantially better than that in the cold season when evaluated by the POD, FPR and CSI indices. However, the spatial and temporal performance of SPPs showed little difference in the A’ index.”
Main conclusions of Sections 3.3 are added in Lines 448-452, “Overall, except RE, the evaluation of all the other four series indices showed the CHIRPS and MSWEP outperformed other products in measuring monthly precipitation in the QTP. Only a few SPPs could capture the precipitation series, with the mean NSE of all stations >0.50, including the CHIRPS, MSWEP and PDIR-Now. The performance of CHIRPS and MSWEP was generally better in the eastern QTP than that at western regions.”

11. L441-L442. The statement "Previous comprehensive evaluation results of SPPs were usually subjectively affected" seems somewhat subjective. Does the author have sufficient evidence to support this argument?
Reply: As mentioned in replies to Comment 2, there are many studies quantitatively evaluated SPPs by each index. However, the comprehensive evaluation results were generally obtained qualitatively. For example, the model 1 might showed the best performance in the A, B, and C evaluation indices, while the model 2 was the best in D and E indices. In these evaluation based on each index, comprehensive evaluation results might be subjectively made that the best model was the model 1 or model 2. The comprehensive evaluation results of the studies in the Introduction are obtained in this way.

12. L485-L466. The author mentions that "This study tried to compare the evaluation results between the long and short series stations and did not find any obvious differences". Evidences should be provided.
Reply: This conclusion is based on the results of several evaluation indices. This study tried to compare the evaluation results between the long and short series stations and did not find any obvious differences. However, the author finds it difficult to provide systematic proof of this conclusion. This sentence “This study tried to compare the evaluation results between the long and short series stations and did not find any obvious differences.” has been deleted in the revised manuscript. 

Comments on the Quality of English Language
Many words are inappropriate. For example, L25, the westerly areas.
Reply: In Line 26 (the revised manuscript), the “westerly areas” has been changed to “westerly circulation areas”. 
In Line 87, the “centre” has been change to “center”.
In Line 573, the “rank score (RS)” has been changed to “RS”
In Line 626, the “centre” has been change to “center”. 

Reviewer 2 Report (New Reviewer)

The provided paper evaluates satellite precipitation products (SPPs) using multiple indices in the data-scarce Qinghai-Tibetan Plateau (QTP). The paper introduces an improved rank score (RS) method and compares ten high-resolution SPPs based on precipitation occurrence and series indices. The evaluation is conducted using a new observation network comparable to the number of national basic stations, aiming to obtain more reliable results. This paper is well written and easy to follow. The study results are supported by data analysis. I have two larger comments and a few minor comments:

1. In the section "Performance of SPPs in measuring precipitation series", we also want to see their performance across different seasons and varying precipitation intensities."

2. The satellite precipitation products used in this study have different spatial resolutions. How does this variability affect the accuracy of the assessments?

3. What is the impact of convective cloud systems in the Qinghai-Tibet Plateau on satellite precipitation products?

4. The article did not specify where the Root Mean Square Error (RMSE) was utilized in the methodology.

5. Please provide information on how to access the remote sensing precipitation data used in the study.

6. The final calculation method for Remote Sensing (RS) was not provided in the manuscript.

Author Response

Responses to Reviewer’ Comments

Manuscript ID: remotesensing-2448871

Title: Comprehensive evaluation of high-resolution satellite precipitation products over the Qinghai-Tibetan Plateau using the new ground observation network

Author(s): Zhaofei Liu

It would be greatly appreciated for your kind reviewing to this paper. The manuscript has been thoroughly revised according to your valuable comments and suggestion. The revised manuscript should, after incorporating your valuable advices and suggestions, be improved greatly. For your convenience to re-review the paper, the response corresponding to your comments are described in detail as follows:

Comments and Suggestions for Authors

The provided paper evaluates satellite precipitation products (SPPs) using multiple indices in the data-scarce Qinghai-Tibetan Plateau (QTP). The paper introduces an improved rank score (RS) method and compares ten high-resolution SPPs based on precipitation occurrence and series indices. The evaluation is conducted using a new observation network comparable to the number of national basic stations, aiming to obtain more reliable results. This paper is well written and easy to follow. The study results are supported by data analysis. I have two larger comments and a few minor comments:

  1. In the section "Performance of SPPs in measuring precipitation series", we also want to see their performance across different seasons and varying precipitation intensities."

Reply: Thanks very much for your valuable comments. The performance of SPPs in detecting precipitation occurrence was analyzed for different periods and precipitation intensities in Section 3.1 and 3.2. This is because the results of these sections were calculated from equations (1) to (5) based on daily series. The calculation results of these indices could be divided into different periods and precipitation intensities. However, the results in the section 3.3 “Performance of SPPs in measuring precipitation series were calculated from equations (6) to (11) based on monthly precipitation series. The monthly series itself is as an entire one, which already contains information about seasons and intensities. In other words, the monthly series is calculated as a whole one for each index, and is difficult to separated it into different periods. Therefore, the performance variations in different seasons and precipitation intensities was only analyzed in section 3.1 and 3.2.

  1. The satellite precipitation products used in this study have different spatial resolutions. How does this variability affect the accuracy of the assessments?

Reply: Thanks very much for your valuable comments. Yes, the temporal and spatial resolutions of satellite precipitation products do not match exactly. This study evaluated these products using ground observations at site scale. The closest pixel to each ground observation site was selected for each product. However, the mismatch between site observation scale and grid satellite product scale is out the scope of this study. According to your valuable comments, these descriptions have been added at the end of section 2.2.3 (Lines 212-215), as follows,

“Ten SPPs were evaluated using the gauge-observed precipitation data from QTP and the raw spatial resolution of each SPP was used for the evaluation. The pixels closest to each ground station were selected for each SPP. The evaluation was based on a daily time series, and evaluation period was the time of intersection of each station and SPP.”

  1. What is the impact of convective cloud systems in the Qinghai-Tibet Plateau on satellite precipitation products?

Reply: The impact of convective cloud systems in the Qinghai-Tibet Plateau on satellite precipitation products is complex, and is out of the scope of this manuscript. However, the author has read some related papers. In Lines 565-568 (the section of “4. Discussions”), several sentences “The QTP is a relatively active area for convective clouds and mesoscale convective systems [48]. Therefore, precipitation processes on the QTP is complex, especially for convective cloud precipitation [49]. This increases the difficulty of satellite precipitation estimates.” have been added to discuss this issue.

  1. The article did not specify where the Root Mean Square Error (RMSE) was utilized in the methodology.

Reply: Thanks very much for your valuable comments. Ten indices were used in this study, but did not include the RMSE. The equation (7) of the RMSE has been deleted in the revised manuscript.

  1. Please provide information on how to access the remote sensing precipitation data used in the study.

Reply: Thanks for your valuable comments. The CHIRPS, CMORPH, GSMaP, IMERG, MSWEP, and PERSIANN products could be downloaded from https://chc.ucsb.edu/data/chirps, https://www.ncei.noaa.gov/products/climate-data-records/precipitation-cmorph, https://developers.google.cn/earth-engine/datasets/catalog, https://disc.gsfc.nasa.gov/datasets/, http://www.gloh2o.org/mswep/, and http://chrsdata.eng.uci.edu/, respectively. These have been added in the section 2.1.2. Please find details in the revised manuscript.

  1. The final calculation method for Remote Sensing (RS) was not provided in the manuscript.

Reply: Thanks very much for your valuable comments. In Lines 206-211, a new paragraph “The RS was first calculated from the mean value of each index for each SPP. The total RS for a SPP was obtained by averaging the RS for each index used. The performance of SPPs in detecting precipitation occurrence and measuring precipitation series was evaluated by the occurrence (POD, FAR, FPR, CSI, and A’) and series indices (RE, CC, IOA, KGE, and NSE), respectively. The SPPs measurements were comprehensively evaluated with the RS of all 10 indices.” has been added to describe the final calculation method.

Reviewer 3 Report (New Reviewer)

Thank you for allowing me to review this article

I have reviewed the manuscript titled "Comprehensive evaluation of high-resolution satellite precipitation products over the Qinghai-Tibetan Plateau using the new ground observation network" which has been submitted to Remote Sensing for potential publication. The study conducted by the author involves the assessment of ten high-resolution satellite precipitation products using an enhanced rank score (RS) approach in the data-limited Qinghai-Tibetan Plateau (QTP). Based on my evaluation, I believe that the topic of this manuscript needs major conceptual revision before being  appropriate for Remote Sensing.

I have concerns about the used RS score. Traditional methods are available and listed by the author which can support in the ranking of the satellite data accuracy.

I think the following procedure can be followed:

categorical metrics (A’, CSI, FAR, FPR, POD) describe the event capturing capability, can be used as a primary ranking indicator prior proceeding to the quantitative statistical metrics. The quantitative statistical metrics can be used also in further two steps the first on using the KGE or NSE. I recommend KGE over the NSE because it provides wider ranges (https://doi.org/10.3390/w15010092). Then finally the satellite data can be evaluated based on other quantitative statistical metrics (CC, SD, CRSME).

Author Response

Responses to Reviewer’ Comments

Manuscript ID: remotesensing-2448871

Title: Comprehensive evaluation of high-resolution satellite precipitation products over the Qinghai-Tibetan Plateau using the new ground observation network

Author(s): Zhaofei Liu

It would be greatly appreciated for your kind reviewing to this paper. The manuscript has been thoroughly revised according to your valuable comments and suggestion. The revised manuscript should, after incorporating your valuable advices and suggestions, be improved greatly. For your convenience to re-review the paper, the response corresponding to your comments are described in detail as follows:

Comments and Suggestions for Authors

I have reviewed the manuscript titled "Comprehensive evaluation of high-resolution satellite precipitation products over the Qinghai-Tibetan Plateau using the new ground observation network" which has been submitted to Remote Sensing for potential publication. The study conducted by the author involves the assessment of ten high-resolution satellite precipitation products using an enhanced rank score (RS) approach in the data-limited Qinghai-Tibetan Plateau (QTP). Based on my evaluation, I believe that the topic of this manuscript needs major conceptual revision before being appropriate for Remote Sensing.

I have concerns about the used RS score. Traditional methods are available and listed by the author which can support in the ranking of the satellite data accuracy.

I think the following procedure can be followed:

categorical metrics (A’, CSI, FAR, FPR, POD) describe the event capturing capability, can be used as a primary ranking indicator prior proceeding to the quantitative statistical metrics. The quantitative statistical metrics can be used also in further two steps the first on using the KGE or NSE. I recommend KGE over the NSE because it provides wider ranges (https://doi.org/10.3390/w15010092). Then finally the satellite data can be evaluated based on other quantitative statistical metrics (CC, SD, CRSME).

Reply: Thanks very much for your valuable comments. Yes, as you mentioned that, traditional methods are the basis for evaluation of satellite precipitation products (SPPs) at regional scale. The procedure of evaluation in this manuscript is consistent with the procedure you suggested. Firstly, the performance of SPPs in capturing precipitation occurrence was analyzed by categorical metrics (A’, CSI, FAR, FPR, POD) in the Section 3.1. In addition, spatial and temporal performance of SPPs was also analyzed in the Section 3.2. Then, the quantitative statistical metrics were applied to evaluate the performance of SPPs in measuring precipitation series, including the KGE, NSE, relative error, CC, and IOA. Yes, the author also found that the KGE and NSE were important indices in evaluating SPPs at regional scales. In addition, the CC, IOA, and relative error were also applied in the evaluation of SPPs in measuring precipitation series. However, the standard deviation (SD) was not used in this study, because it has been included in the KGE. In Lines 171-173, the sentence “The KGE is an evaluation index that integrates CC, RE, and standard deviation.” has been revised to “The KGE value, which integrates CC, RE, and standard deviation (SD), was used instead of a single standard deviation.” to explain the rationality of using KGE together with RE and CC as evaluation metrics instead of the SD. These evaluation results were desbribed in the Section 3.3. Finally, an improved rank score (RS) method was used for the quantitative comprehensive evaluation of SPPs at the study area as described in the Section 3.4. According to your valuable comments, a new reference has been added in the revised manuscript as follows,

Added reference,

  1. Helmi, A.M.; Abdelhamed, M.S. 2023. Evaluation of CMORPH, PERSIANN-CDR, CHIRPS V2.0, TMPA 3B42 V7, and GPM IMERG V6 satellite precipitation datasets in Arabian arid regions. Water 15, 92. https://doi.org/10.3390/w15010092.

Round 2

Reviewer 1 Report (New Reviewer)

In summary, the author's work is substantial, and the response to the review comments is generally satisfied. The following two points should be clarified.

In the previous question (the 1st), "Is it appropriate to compare these products together? For example, PERSIANN CDR aims to provide long-term continuous precipitation observations, so it is expected to have lower precipitation detection and estimation accuracy compared to GPM." I meant that since the quality of GPM is obviously superior to PERSIANN-CDR, why the PERSIANN-CDR was chosen as one among the satellite precipitation products to be compared? An apparent knowledge should not be token as a conclusion in this paper.

In the previous question (the 8th), "It is mentioned how various satellite precipitation products perform in different months, but this aspect is not reflected in Figure 5." I meant that you mentioned the performance of various satellite precipitation products in different months in the paragraphs around Figure 5, but Figure 5 only provides a vague maximum, minimum, and median values. For example, L346-L347 mentioned that "CHIRPS showed the strongest performance with the lowest FAR". However, such conclusions cannot be derived from Figure 5.

It is well.

Author Response

Responses to Reviewer’ Comments

Manuscript ID: remotesensing-2448871

Title: Comprehensive evaluation of high-resolution satellite precipitation products over the Qinghai-Tibetan Plateau using the new ground observation network

Author(s): Zhaofei Liu

It would be greatly appreciated for your kind reviewing to this paper. The manuscript has been revised according to your valuable comments and suggestion. The revised manuscript should, after incorporating your valuable advices and suggestions, be improved greatly. For your convenience to re-review the paper, the response corresponding to your comments are described in detail as follows:

Comments and Suggestions for Authors

In summary, the author's work is substantial, and the response to the review comments is generally satisfied. The following two points should be clarified.

In the previous question (the 1st), "Is it appropriate to compare these products together? For example, PERSIANN CDR aims to provide long-term continuous precipitation observations, so it is expected to have lower precipitation detection and estimation accuracy compared to GPM." I meant that since the quality of GPM is obviously superior to PERSIANN-CDR, why the PERSIANN-CDR was chosen as one among the satellite precipitation products to be compared? An apparent knowledge should not be token as a conclusion in this paper.

Reply: Thanks very much for your valuable comments. In general, some satellite precipitation products (SPPs) (e.g. PERSIANN-CDR) might be expected to have lower performance compared to other products (e.g. GPM). However, this study not only evaluates the rank of performance among SPPs, but also evaluates the specific performance of each product. For example, although the overall performance of the product A is better than that of the product B, the product B might outperform the product A in some evaluation indices. It is also possible that both of the product A and B perform well, for example, with correlation coefficients > 0.80. The difference is that the value of the correlation coefficient despite is 0.86 and 0.82, respectively. The PERSIANN CDR and GPM products also meet these characteristics. As shown in Table 2, although the overall performance of PERSIANN CDR is lower than that of GPM, the former outperforms the latter in the index of POD, CC, and IOA.

In addition, several sentences had been added in Lines 548-552 for discussions, “It should be noted that in addition to the variations in accuracy performance, the SPPs evaluated in this study also have their own advantages. For example, PERSIANN CDR aims to provide long-term continuous precipitation estimates, so it is expected to have lower accuracy compared to GPM. Therefore, it is suggested that these issues could be considered in the application of SPPs.”

In the previous question (the 8th), "It is mentioned how various satellite precipitation products perform in different months, but this aspect is not reflected in Figure 5." I meant that you mentioned the performance of various satellite precipitation products in different months in the paragraphs around Figure 5, but Figure 5 only provides a vague maximum, minimum, and median values. For example, L346-L347 mentioned that "CHIRPS showed the strongest performance with the lowest FAR". However, such conclusions cannot be derived from Figure 5.

Reply: Thanks very much for your valuable comments. A new table “Table S2. Performance of each SPPs in detecting precipitation occurrence during different periods” has been added in “Supplementary Materials” to support these conclusions.

Table S2. Performance of each SPPs in detecting precipitation occurrence during different periods

SPPs

Periods

CMO

GPM

MS

PER

CHI

GaM

GaN

PDI

PCS

PDR

POD

Jan

0.19

0.05

0.77

0.69

0.39

0.17

0.15

0.40

0.74

0.61

Feb

0.31

0.08

0.80

0.71

0.46

0.19

0.20

0.51

0.75

0.68

Mar

0.47

0.16

0.84

0.75

0.49

0.36

0.33

0.69

0.79

0.75

Apr

0.64

0.27

0.88

0.75

0.44

0.54

0.51

0.81

0.76

0.82

May

0.68

0.41

0.94

0.61

0.26

0.65

0.61

0.83

0.59

0.80

Jun

0.69

0.53

0.97

0.51

0.30

0.71

0.68

0.80

0.50

0.76

Jul

0.79

0.69

0.98

0.61

0.44

0.77

0.74

0.84

0.60

0.83

Aug

0.80

0.66

0.98

0.60

0.42

0.77

0.74

0.82

0.59

0.82

Sep

0.71

0.52

0.97

0.47

0.30

0.62

0.63

0.78

0.46

0.76

Oct

0.60

0.41

0.89

0.45

0.21

0.44

0.46

0.73

0.48

0.67

Nov

0.42

0.20

0.78

0.55

0.28

0.19

0.25

0.62

0.60

0.63

Dec

0.24

0.07

0.69

0.58

0.32

0.14

0.14

0.41

0.63

0.47

Warm

0.77

0.62

0.98

0.57

0.38

0.74

0.71

0.83

0.56

0.81

Cold

0.33

0.11

0.78

0.69

0.45

0.27

0.24

0.54

0.75

0.67

Annual

0.70

0.53

0.96

0.61

0.38

0.64

0.62

0.79

0.60

0.79

FAR

Jan

0.23

0.01

0.24

0.42

0.14

0.10

0.08

0.20

0.44

0.34

Feb

0.36

0.03

0.30

0.50

0.21

0.12

0.10

0.30

0.52

0.42

Mar

0.44

0.06

0.36

0.54

0.26

0.23

0.20

0.46

0.56

0.57

Apr

0.54

0.12

0.47

0.51

0.21

0.34

0.30

0.54

0.49

0.59

May

0.53

0.12

0.56

0.36

0.11

0.44

0.39

0.55

0.34

0.59

Jun

0.43

0.19

0.71

0.24

0.12

0.45

0.38

0.50

0.25

0.52

Jul

0.42

0.30

0.77

0.30

0.19

0.44

0.36

0.50

0.30

0.57

Aug

0.43

0.24

0.77

0.25

0.15

0.43

0.35

0.44

0.25

0.51

Sep

0.37

0.15

0.67

0.16

0.09

0.34

0.28

0.37

0.15

0.39

Oct

0.46

0.13

0.48

0.18

0.06

0.19

0.20

0.33

0.19

0.33

Nov

0.40

0.05

0.27

0.25

0.06

0.10

0.13

0.27

0.26

0.28

Dec

0.28

0.02

0.19

0.29

0.09

0.06

0.07

0.15

0.30

0.22

Cold

0.40

0.21

0.72

0.22

0.13

0.40

0.33

0.43

0.22

0.48

Warm

0.32

0.03

0.27

0.43

0.17

0.12

0.11

0.26

0.44

0.37

Annual

0.38

0.12

0.47

0.34

0.14

0.26

0.23

0.37

0.33

0.44

FPR

Jan

0.83

0.66

0.63

0.76

0.71

0.87

0.80

0.79

0.80

0.79

Feb

0.82

0.56

0.62

0.74

0.68

0.78

0.72

0.75

0.75

0.75

Mar

0.72

0.44

0.54

0.68

0.62

0.66

0.65

0.67

0.68

0.69

Apr

0.59

0.35

0.46

0.57

0.50

0.56

0.52

0.56

0.55

0.57

May

0.46

0.26

0.38

0.43

0.37

0.47

0.41

0.45

0.42

0.46

Jun

0.32

0.23

0.34

0.31

0.26

0.37

0.30

0.35

0.31

0.36

Jul

0.22

0.18

0.27

0.21

0.17

0.29

0.21

0.24

0.21

0.27

Aug

0.25

0.20

0.31

0.23

0.20

0.32

0.24

0.27

0.23

0.30

Sep

0.35

0.24

0.38

0.31

0.28

0.44

0.33

0.35

0.30

0.36

Oct

0.62

0.46

0.53

0.53

0.46

0.64

0.53

0.55

0.52

0.56

Nov

0.81

0.66

0.66

0.74

0.66

0.84

0.79

0.77

0.77

0.78

Dec

0.84

0.75

0.67

0.77

0.79

0.89

0.88

0.83

0.85

0.84

Cold

0.27

0.20

0.32

0.25

0.21

0.34

0.25

0.29

0.25

0.31

Warm

0.77

0.56

0.59

0.73

0.67

0.74

0.73

0.74

0.75

0.75

Annual

0.49

0.25

0.40

0.45

0.39

0.43

0.36

0.43

0.47

0.47

CSI

Jan

0.08

0.04

0.32

0.16

0.16

0.07

0.08

0.14

0.17

0.17

Feb

0.11

0.06

0.35

0.19

0.19

0.10

0.12

0.19

0.20

0.21

Mar

0.21

0.14

0.43

0.25

0.23

0.19

0.19

0.28

0.27

0.28

Apr

0.34

0.23

0.50

0.35

0.27

0.34

0.34

0.39

0.37

0.39

May

0.44

0.35

0.60

0.40

0.22

0.42

0.43

0.49

0.39

0.46

Jun

0.53

0.45

0.65

0.41

0.26

0.51

0.54

0.57

0.41

0.53

Jul

0.65

0.59

0.72

0.52

0.40

0.59

0.63

0.66

0.51

0.64

Aug

0.63

0.56

0.68

0.49

0.38

0.56

0.60

0.62

0.49

0.60

Sep

0.51

0.44

0.61

0.38

0.26

0.43

0.49

0.55

0.38

0.52

Oct

0.30

0.29

0.45

0.27

0.16

0.25

0.31

0.37

0.28

0.33

Nov

0.13

0.12

0.29

0.15

0.15

0.08

0.11

0.18

0.17

0.18

Dec

0.08

0.04

0.24

0.12

0.12

0.05

0.05

0.11

0.12

0.12

Cold

0.60

0.53

0.67

0.47

0.34

0.54

0.58

0.61

0.47

0.59

Warm

0.14

0.09

0.35

0.20

0.19

0.13

0.13

0.21

0.20

0.22

Annual

0.44

0.44

0.58

0.37

0.29

0.44

0.46

0.48

0.37

0.46

Accuracy

Jan

0.67

0.81

0.75

0.55

0.80

0.83

0.83

0.76

0.58

0.66

Feb

0.59

0.79

0.73

0.52

0.73

0.79

0.78

0.69

0.52

0.61

Mar

0.58

0.76

0.74

0.51

0.67

0.69

0.70

0.60

0.51

0.54

Apr

0.59

0.70

0.71

0.57

0.64

0.68

0.69

0.60

0.58

0.56

May

0.62

0.67

0.72

0.62

0.59

0.67

0.67

0.64

0.62

0.59

Jun

0.69

0.67

0.72

0.63

0.55

0.69

0.71

0.69

0.62

0.64

Jul

0.73

0.71

0.76

0.67

0.59

0.71

0.73

0.74

0.65

0.72

Aug

0.73

0.70

0.73

0.66

0.60

0.70

0.72

0.72

0.66

0.70

Sep

0.70

0.69

0.70

0.67

0.59

0.69

0.71

0.71

0.66

0.69

Oct

0.59

0.74

0.69

0.70

0.72

0.77

0.74

0.69

0.71

0.67

Nov

0.58

0.80

0.74

0.68

0.85

0.84

0.80

0.72

0.72

0.71

Dec

0.65

0.82

0.76

0.63

0.86

0.88

0.86

0.81

0.69

0.76

Warm

0.71

0.69

0.73

0.66

0.58

0.70

0.72

0.72

0.65

0.69

Cold

0.62

0.79

0.74

0.56

0.77

0.79

0.79

0.72

0.58

0.64

Annual

0.67

0.75

0.74

0.63

0.67

0.74

0.74

0.70

0.63

0.66

* Note: The underline means the worst performance, while bold font represents the best performance.

Comments on the Quality of English Language

It is well.

Reply: Thanks very much.

Reviewer 2 Report (New Reviewer)

All my concerns have been addressed.

Author Response

Thanks very much.

Reviewer 3 Report (New Reviewer)

Previous comments g

Author Response

Responses to Reviewer 3’ Comments

Manuscript ID: remotesensing-2448871

Title: Comprehensive evaluation of high-resolution satellite precipitation products over the Qinghai-Tibetan Plateau using the new ground observation network

Author(s): Zhaofei Liu

It would be greatly appreciated for your kind reviewing to this paper. The manuscript has been thoroughly revised according to your valuable comments and suggestion. The revised manuscript should, after incorporating your valuable advices and suggestions, be improved greatly. For your convenience to re-review the paper, the response corresponding to your comments are described in detail as follows:

Comments and Suggestions for Authors

Previous comments g

The follows are previous Comments and Suggestions for Authors

I have reviewed the manuscript titled "Comprehensive evaluation of high-resolution satellite precipitation products over the Qinghai-Tibetan Plateau using the new ground observation network" which has been submitted to Remote Sensing for potential publication. The study conducted by the author involves the assessment of ten high-resolution satellite precipitation products using an enhanced rank score (RS) approach in the data-limited Qinghai-Tibetan Plateau (QTP). Based on my evaluation, I believe that the topic of this manuscript needs major conceptual revision before being appropriate for Remote Sensing.

I have concerns about the used RS score. Traditional methods are available and listed by the author which can support in the ranking of the satellite data accuracy.

I think the following procedure can be followed:

categorical metrics (A’, CSI, FAR, FPR, POD) describe the event capturing capability, can be used as a primary ranking indicator prior proceeding to the quantitative statistical metrics. The quantitative statistical metrics can be used also in further two steps the first on using the KGE or NSE. I recommend KGE over the NSE because it provides wider ranges (https://doi.org/10.3390/w15010092). Then finally the satellite data can be evaluated based on other quantitative statistical metrics (CC, SD, CRSME).

Reply: Thanks for your comments. This study focus on Comprehensive evaluation of high-resolution satellite precipitation products over the Qinghai-Tibetan Plateau using the new ground observation network. As mentioned in the manuscript, “Satellite precipitation products (SPPs) have been widely evaluated at regional scales. However, there was little quantitative comprehensive evaluations of SPPs using multiple indices.” There are many studies quantitatively evaluated SPPs by each index. However, the comprehensive evaluation results were generally obtained qualitatively. For example, the model 1 might showed the best performance in the A, B, and C evaluation indices, while the model 2 was the best in D and E indices. In these evaluation based on each index, comprehensive evaluation results might be subjectively made that the best model was the model 1 or model 2. In this study, an improved rank score (RS) method was developed for the quantitative comprehensive evaluation of SPPs. The proposed improved RS method can also potentially be applied for comprehensive evaluation of other products and models. In addition, as for the study area, both the new observation network and national basic stations were used to obtain reliable comprehensive evaluation results for the data scarce Qinghai-Tibetan Plateau.

Yes, as you mentioned that, traditional methods are the basis for evaluation of satellite precipitation products (SPPs) at regional scale. The procedure of evaluation in this manuscript is consistent with the procedure you suggested. Firstly, the performance of SPPs in capturing precipitation occurrence was analyzed by categorical metrics (A’, CSI, FAR, FPR, POD) in the Section 3.1. In addition, spatial and temporal performance of SPPs was also analyzed in the Section 3.2. Then, the quantitative statistical metrics were applied to evaluate the performance of SPPs in measuring precipitation series, including the KGE, NSE, relative error, CC, and IOA. Yes, the author also found that the KGE and NSE were important indices in evaluating SPPs at regional scales. In addition, the CC, IOA, and relative error were also applied in the evaluation of SPPs in measuring precipitation series. However, the standard deviation (SD) was not used in this study, because it has been included in the KGE. In Lines 171-173, the sentence “The KGE is an evaluation index that integrates CC, RE, and standard deviation.” had been revised to “The KGE value, which integrates CC, RE, and standard deviation (SD), was used instead of a single standard deviation.” to explain the rationality of using KGE together with RE and CC as evaluation metrics instead of the SD. These evaluation results were described in the Section 3.3. Finally, an improved rank score (RS) method was used for the quantitative comprehensive evaluation of SPPs at the study area as described in the Section 3.4.

This manuscript is a resubmission of an earlier submission. The following is a list of the peer review reports and author responses from that submission.

Round 1

Reviewer 1 Report

I am afraid that after the first round of revision, my concern remains. I do not think the proposed RS method that integrates all the 10 indices makes any sense. How to interpret the RS results? What does the RS value mean? The authors still failed to answer and response to these questions. I therefore would debut about the innovation and reliability of this work.

The English language is overall fine.

Reviewer 2 Report

This study provides a comprehensive evaluation of ten satellite precipitation products (SPPs) over the Qinghai-Tibetan Plateau using rain gauge data. The main analysis is based on the comparisons among various statistic metrics. Publication of this study can partially mitigate the lack of comprehensive understanding of SPPs over Qinghai-Tibetan Plateau due to complex topography and scarcity of ground observations.

The overall description is pretty good. However, the captions for almost all figures and tables are too simplified.

I had a hard time to understand Fig. 3 even after I read the entire manuscript.   Fig 3. only shows performance of overall SPPs w.r.t months.  I don’t see any info about precipitation levels for each product while the author talked this figure a lot about the performance of SPPs in detecting precipitation occurrence at different levels of precipitation. I even doubt the author inputted a wrong figure here.

As for Fig.2, I’d like to suggest to specifically indicate data range and resolution (3-h?, 0.1deg?).

The descriptions for some SPP products in Table 1 are inaccurate and inconsistent. For example, GPM is a satellite mission, instead of a product. IMERG-Final is one of the GPM products, instead of a version. Latest IMERG is Version06B as of today. The author indicated the version of IMERG in text but incorrectly labeled it in the table. IMERG is at 30-minute resolution, rather than 3-hour.  Please check all other products and correct the descriptions.

Lines 500-502. “Previous comprehensive evaluation results of SPPs were usually obtained from qualitative analyses of multiple indexes”.  There  are many publications quantitatively evaluated SPPs.

Line 150 "confusion matrix"? Does the author mean the contingency table?

Line 204 ACC (A’,  accuracy) is not defined, which may need to be defined in Line 154.

Line 500. "This study can chiefly be improved……"  I don't understand what the author is trying to express.

The overall quality of English in this paper is pretty good though I do find some instances which may be related to language issue as pointed above.

Reviewer 3 Report

Dear Editor,

Please find my review of a manuscript remotesensing- 2375902 "Comprehensive evaluation of high-resolution satellite precipitation products over the Qinghai-Tibetan Plateau using the new ground observation network" by Zhaofei Liu submitted to Remote Sensing for consideration for possible publication.

In this study, the author evaluated ten high-resolution satellite precipitation products using an improved rank score (RS) method in the data-scarce Qinghai-Tibetan Plateau (QTP). The subject of this report is suitable for Remote Sensing and it could be published after suggested revision.

Lines 39 – 40: Gauge observation is the most intuitive method for precipitation measurement [3].

Please reconsider this statement. Perhaps replace with something like "Gauge observation is considered as the most accurate method for precipitation measurement ".

Lines 40 – 41: However, most precipitation gauge stations are concentrated in the eastern China.

Please consider re-wording, e.g.,  "However, most precipitation gauge stations in China are concentrated in the eastern part of the country."

Lines 48 and 50: Satellite measurements … Recommend replacing "measurements" with "precipitation estimates".

Line 51: In recent times, the satellite-based precipitation products are becoming

=> An acronym for SPPs should be introduced here. In recent times, the satellite-based precipitation products (SPPs) are becoming

 Lines 54 – 55: However, satellite-based precipitation products (SPPs) have inherent drawbacks associated with the indirect nature of the

=> Please delete (SPPs). However, satellite-based precipitation products have inherent drawbacks associated with the indirect nature of the

Lines 54  - 59: Please remove highlighting (the same for lines 83 - 91).

Lines 60 - 64:   Evaluation of SPPs showed that the integrated multi-satellite retrieval (IMERGs) products from the global precipitation measurements (GPMs) generally exhibited the strongest performance compared to other products at global and region scales [10]. These included Australia [11], Bangladesh [12], Central Asia [13], Eastern Himalayas [7], India [14], Nigeria [15], Thailand [16], and Vietnam [17].

Suggest including a reference on more recent than [11] study for Australia as well as a study for the Pacific.

Wild, A., Chua, Z-W., Kuleshov, Y., 2022: Triple Collocation Analysis of Satellite Precipitation Estimates over Australia.   Remote Sensing. 2022, 14, 2724. https://doi.org/10.3390/rs14112724.

T. Tashima, T. Kubota, T. Mega, T. Ushio and R. Oki, "Precipitation Extremes Monitoring Using the Near-Real-Time GSMaP Product," in IEEE Journal of Selected Topics in Applied Earth Observations and Remote Sensing, vol. 13, pp. 5640-5651, 2020, doi: 10.1109/JSTARS.2020.3014881.

Lines 257 – 258: Figure 3. Performance of satellite products in detecting precipitation occurrence at different levels of precipitation.

Please ensure that the figure caption is on the same page as Figure 3.

Line 600: 10 indexes representing precipitation occurrence and time series by an improved rank

=> suggest replacing "10 indexes" with "ten indices "

This reviewer recommends accepting the manuscript for publication after a minor revision.

Yours faithfully,

The Reviewer

The manuscript is well written, only minor editing of English language is required. 

Reviewer 4 Report

This manuscript validates ten satellite-based precipitation products over the Qinghai-Tibetan Plateau using ground observations. The authors rank score based method to evaluate the satellite-based products. I have several issues with this paper and thus suggest a reject.

1. Writing: Writing definitely needs to be improved. Many repetitive words and grammatical issues make the manuscript difficult to read. Some sentences in the introduction are incorrect. 

2. Novelty: The work lacks novelty. The introduction lacks strong motivation on why the rank score is preferred over other traditional methods. Also, how is this study different from other validation studies?

3. Data: I understand the authors' motivation to consider all products of PERSIANN. But, why all products of IMERG is not considered for validation. 

4. Methods: 

a. Definition of false positives and false negatives is incorrect. 

b. At what timescales are these validation conducted? Most products have data at daily and sub-daily timescales. 

c. Line 204: What is ACC

d. The major problem with methods - I fail to understand why Rank Score is even needed. This is just a normalization of the existing metrics. 

5. Results: 

a. Figure 4 - I don't understand the use of averaging metrics across all products. Same for Figure 5, Figure 7. 

b. Table 2: How do the authors derive RSC, RSM and RSO? 

c. The interpretation of results needs to be rewritten. 

Extensive english editing is needed. 

For example:

Line 12: "multiple indexes"?

Line 19: Why concentrated in certain products?

Line 27-28

Line 20: What is RE

Line 57: Inaccuracies are not due to low gauges. that is due to complexity in the processes, sensor sensitivities and so on. 

Line 67, Line 69, Line 200.... 

I listed only few examples. The results and discussion needs to improved too!